# Insights into HP1a-Chromatin Interactions

**DOI:** 10.3390/cells9081866

**Published:** 2020-08-09

**Authors:** Silvia Meyer-Nava, Victor E. Nieto-Caballero, Mario Zurita, Viviana Valadez-Graham

**Affiliations:** Instituto de Biotecnología, Departamento de Genética del Desarrollo y Fisiología Molecular, Universidad Nacional Autónoma de México, Cuernavaca Morelos 62210, Mexico; smeyer@ibt.unam.mx (S.M.-N.); vnieto@lcg.unam.mx (V.E.N.-C.); marioz@ibt.unam.mx (M.Z.)

**Keywords:** heterochromatin, HP1a, genome stability

## Abstract

Understanding the packaging of DNA into chromatin has become a crucial aspect in the study of gene regulatory mechanisms. Heterochromatin establishment and maintenance dynamics have emerged as some of the main features involved in genome stability, cellular development, and diseases. The most extensively studied heterochromatin protein is HP1a. This protein has two main domains, namely the chromoshadow and the chromodomain, separated by a hinge region. Over the years, several works have taken on the task of identifying HP1a partners using different strategies. In this review, we focus on describing these interactions and the possible complexes and subcomplexes associated with this critical protein. Characterization of these complexes will help us to clearly understand the implications of the interactions of HP1a in heterochromatin maintenance, heterochromatin dynamics, and heterochromatin’s direct relationship to gene regulation and chromatin organization.

## 1. Introduction

Chromatin is a complex of DNA and associated proteins in which the genetic material is packed in the interior of the nucleus of eukaryotic cells [1]. To organize this highly compact structure, two categories of proteins are needed: histones [2] and accessory proteins, such as chromatin regulators and histone-modifying proteins. Both kinds of proteins participate in maintaining the structure of chromatin and regulating gene expression [3]. The primary unit of chromatin is the nucleosome [4], which is formed by an octamer of histones, with two copies of histones H2A, H2B, H3, and H4 (also called the canonical or core histones) [4]. The histone H1 has been referred to as a “linker” because a single copy is positioned on the DNA between each nucleosome [5]. Deciphering the procedures that control chromatin packaging has become a significant issue in understanding developmental programs and disease states.

There are two primary types of chromatin in the nucleus: heterochromatin and euchromatin [6]. Heterochromatin is abundant in compacted, highly condensed, silenced, and repetitious sequences found near centromeric and telomeric locations. By contrast, euchromatin includes the majority of transcriptionally active genes [7]. Through staining different types of cells, Emil Heitz conceived the term “heterochromatin” more than 90 years ago, observing retention of this more compact structure throughout the cell cycle [8]. These core heterochromatic structures have become an essential area of study because of their role in gene silencing [9].

In all eukaryotes, constitutive heterochromatin is established early in development. During the 1960s, satellite sequences were identified, sequenced, and mapped to pericentromeric and telomeric regions of metaphase chromosomes located at the nuclear periphery of interphase cells [10]. With the development of automatic sequencing over the decades that followed, studies on vertebrates have determined that the genome is rich in repetitive sequences that, for example, account for more than 50% of the human genome. There are many types of these repetitive elements: some are composed of retrotransposon sequences, others of long and short interspersed elements known as LINEs, SINEs, Alu sequences, in addition to minor and major satellite sequences. These sequences need to be silenced to avoid chromosome instability, and several mechanisms cooperate toward maintaining this silencing. These mechanisms include DNA methylation, histone post-transcriptional modifications, histone deacetylation, binding of chromatin proteins, and non-coding RNA and RNA interference pathways [11,12,13]. Embryonic stem cells have, in general, less heterochromatin than differentiated cells. This characteristic confers plasticity. As differentiation advances, cells gain heterochromatin. Disruption of any of these heterochromatin maintenance mechanisms leads to chromosome instability and can sometimes lead to diseases such as cancer.

The mechanisms of heterochromatin formation and maintenance have been highly conserved throughout the evolution of eukaryotic cells, and understanding these mechanisms using less complex animal models has helped us to advance understanding in this important field.

Based on cytological criteria, one-third of the *Drosophila melanogaster* genome, including the telomeres, pericentric regions, and chromosome 4, is considered as the heterochromatin [14]. As development and differentiation progress, regions regarded as heterochromatin become more abundant as differentiated cells undergo heterochromatinization to promote gene repression and prevent inappropriate gene expression. One mechanism for achieving this is for cells to anchor chromatin to the nuclear lamina resulting in gene inactivation [15]; alternatively, the heterochromatin/euchromatin borders may be defined [16], for example, by changing the profile of chromatin as differentiation progresses, i.e., as stem cells differentiate into the mature cell type [17].

The primary mechanism used to maintain differential expression patterns is the silencing of genes, which involves packaging them in structures inaccessible to DNA-binding proteins [18]. The silencing of a specific gene or chromosomal region requires covalent modification by enzymes or complexes harboring subunits that recognize these modifications and facilitate their physical association with histones [19] and their extension throughout the chromatin fiber, creating a compacted structure (heterochromatin) which is generally believed to be inaccessible to transcription-promoting factors [20]. Heterochromatinization then becomes one of the primary mechanisms used to silence chromosomal regions.

In 1930, experiments using X-ray treatment of flies have shown that genes that were translocated from euchromatic regions to the vicinity of pericentric heterochromatin, acquired a motley pattern of expression [21]. This effect, which is caused by the repressive properties of heterochromatin, was called position effect variegation (PEV) and has been exploited from the 1980s onward for the systematic examination of factors that regulate heterochromatin formation. One of the proteins identified through this screening is heterochromatin protein 1 (HP1). It is a highly conserved protein [22] that was initially discovered in *Drosophila* by the group of Grigliatti in a study in which the authors found more than 50 loci that acted as suppressors of PEV. The authors identified that the protein encoded by the *Su(var)2-5* locus works as a dosage-dependent modifier of PEV [23]. Since then, various studies have shown that this protein is essential for the establishment and maintenance of heterochromatin.

HP1 proteins are conserved in a variety of organisms, including fission yeast (as Swi6 and Chp2) [24,25] and also vertebrates such as amphibians (e.g., frog (xHP1α and xHP1γ)) [26], birds (e.g., chicken (HP1α, HP1β, and HP1γ)) [27], and mammals (such as mice (HP1α, HP1β, and HP1γ)) [28]. Various functions have been described for each member of the family throughout the life cycle of a cell: heterochromatin formation and maintenance, gene silencing, telomere capping, DNA repair, and control of gene expression [14]. Mutations that affect HP1 protein activities have a significant impact on organism development. For example, in *Drosophila*, null mutants for HP1a are lethal at the embryonic stage [29]. Although the HP1 isoforms are very similar structurally, they have different functions, and null mutants for HP1a cannot be rescued by HP1b or HP1c. Thus, HP1 proteins have been revealed to interact with a wide variety of proteins, forming different complexes [30,31,32].

In this review, we present a general overview of HP1 proteins, their conserved domains, and their interactions with other proteins. We focus mainly on HP1a to provide a layered view of its interactions as well as their possible impacts on functions and heterochromatin maintenance.

## 2. Functions of Conserved HP1a Domains

HP1a has two highly conserved domains, the N-terminal chromodomain (CHD) that is located in numerous chromosomal proteins [18] and a C-terminal chromoshadow domain (CSD), which are separated by a hinge region of variable length (Hin). The CHD is found in many chromosomal proteins whose primary function is in the maintenance of chromatin structure and gene regulation [33]. The specificity of the CHD for certain modified histone residues is one of the features that direct the binding of these proteins to specific regions in the chromatin [34,35]. HP1 proteins specifically bind to dimethylated and trimethylated H3K9 (H3K9me2 and H3K9me3) through their CHD. This binding of this histone mark to CHD occurs via the region Gln5 to Ser10. These amino acids form a β-sheet that aligns, antiparallel, with two β-sheets that are formed by the regions Glu23 to Val26 and Asn60 and Asp62 in the chromodomain, thus creating a structure of three β-sheets in the form of a sandwich [36] (Figure 1). The HP1a CHD also interacts with the tail of the linker histone H1.4 that is methylated at lysine 26, resulting in greater compaction of chromatin [37]. HP1 has been considered as a sign of repression because it is mainly found in silenced chromatin. Any null mutations in *Su(var)2-5* (HP1a coding gene) and the replacement of H3K9 with arginine (H3K9R) to block HP1a binding are lethal to the organism [38,39,40]. In *Drosophila* HP1a, a single amino acid substitution within the CHD (V26M) is present in the *Su(var)2-5^02^* allele; for this allele, heterozygous flies show the suppression of gene silencing by heterochromatin [38]. Furthermore, a significant reduction of HP1a occupancy near the centromeres and a decrease in survival until the third larval stage have been shown in flies that have a null allele of *Su(var)2-5* and are trans-heterozygous for *Su(var)2-5^02^* [41]. In agreement with these results, the crucial role of V26 in forming the hydrophobic pocket of CHD that binds to H3K9me has been demonstrated through crystallographic studies [36]. Thus, the CHD is essential for the whole protein to target this heterochromatin mark, and a simple amino acid substitution can be lethal to the organism.

The second domain shares identity with the amino acid sequence of the CHD and was thus named the chromoshadow domain (CSD) [44]. A function critical for the formation of heterochromatin is preserved within this domain [45], which facilitates the dimerization of HP1 proteins and also directs interactions with other proteins that carry the conserved pentapeptide motif, PxVxL (x = any amino acid) (Table 1) [46,47]. The structure of the CSD is roughly similar to that of the CHD (three β-sheets packed against two α-helices) [48]. For example, a single amino acid replacement inside the CSD (I161E) prevents the dimerization of mouse HP1β [33]. The absence of dimerization also triggers the loss of contact with nuclear factors carrying PxVxL motifs as well as non-PxVxL partners [49,50]. By contrast, a single amino acid replacement elsewhere in the CSD (W170A) of mouse HP1β does not preclude dimerization but disturbs interactions with PxVxL partner proteins [33]. Consequently, the binding to PxVxL proteins and the conditions for HP1 dimerization can be eliminated independently.

At first, the hinge region was thought of as being only a linker region [111] because it is the region corresponding to the greatest amino acid variability within HP1 proteins. Moreover, other studies have suggested that its structure is flexible and disorganized [56,57,112,113]. However, the hinge region has been found to contribute to facilitating specific interactions [26,42,47,114,115] and is also highly receptive to subsequent post-translation modifications, especially phosphorylation [112,116,117]. Furthermore, changes within this region were shown to alter the location, interactions, and function of HP1 proteins, thus making it a critical control region in the regulation of HP1 proteins [117,118,119].

Both the CHD and the CSD have been the focus of extensive structural analyses [33,36,48], which have determined that each domain forms a hydrophobic pocket. Recently, using cryogenic electron microscopy, Machida et al. reported the three-dimensional structure of a complex containing dinucleosomes with H3K9me3 modification and human HP1 isoforms. In these structures, two H3K9me3 nucleosomes are joined by a symmetric HP1 dimer (for example, an α with an α). The linker DNA between the nucleosomes does not interact directly with HP1, thereby allowing the nucleosome to be remodeled by ATP-utilizing chromatin assembly factor (ACF) [120]. This is an important observation because it changes the view of heterochromatin from being stable and rigid regions to regions that can also be highly malleable and where diverse cellular mechanisms, such as DNA repair or transcription, can take place.

Just as the CHD is preserved within the protein, the proteins of the HP1 family have been conserved throughout evolution. Most eukaryotes have three primary genes encoding variants of HP1 proteins with different functions. Humans have three principal isoforms of HP1 (referred to as HP1α, HP1β, and HP1γ) [121]. *Drosophila* also expresses three primary isoforms of HP1, encoded by different genes (HP1a, HP1b, and HP1c), which are ubiquitously expressed in adult fly [122] (Figure 2a). Flies also have two germline-specific isoforms, HP1d (Rhino) and HP1e. Rhino is expressed in the ovaries and involved in transposon silencing in the germline via piRNA clusters [123]. HP1e is expressed in the testes and is essential for paternal chromosome segregation through embryonic mitosis [122].

Regarding their functions, the paralogs of this family show considerable differences in location. The *Drosophila* HP1a and mammalian HP1α are predominantly localized to heterochromatin [47,55,62,124]. HP1b (both *Drosophila* and mammalian) is present in heterochromatin and euchromatin, whereas HP1c localizes to euchromatin and yields gene-specific contributions to transcriptional regulation [125]. Given these differences in location, it seems that these paralogs can form different complexes or interact at various places in chromatin.

Although HP1 proteins share high similarity with respect to both their amino acid sequences and their comprising domains, they present differences in the disposition of these domains [44], with many such differences observed between *Drosophila* paralogs [126]. For example, the HP1a CHD is located between amino acids 20 and 80, whereas in HP1b and c, the CHD domain is located almost at the beginning of the protein. The hinge region is shorter in HP1b and c compared to HP1a and is the least conserved region among all *Drosophila* homologs [127]. The hinge region connecting the two main domains seems to enable the CHD and CSD to move independently of each other in the native protein [33].

Further, in vitro studies have shown that phosphorylation of the most N-terminal portion of HP1α inhibits DNA binding but promotes phase separation by creating subcompartments where the same protein can be located in chromatin with distinct grades of compaction [128]. This N-terminal part is almost entirely absent in β and γ (see Figure 2b). Lastly, in the C-terminal part after the CSD domain, HP1a has three amino acids. By comparison, HP1b and HP1c each have a C-terminal extension region (CTE) with a length of 85 and 96 amino acids, respectively. Analysis of these CTE sequences did not reveal any similarities either between them or with any reported domains, and further studies of their contribution to the function of these proteins will be of great importance [53].

Although the CHDs of all three HP1 proteins are involved in the recognition and binding of H3K9me2/3, they do not bind with the same affinity. In competition experiments to test the binding affinity to the trimethylation mark, HP1c always presented the lowest affinity; later, it was confirmed that this mark is not recognized by HP1c in vivo [53]. The overexpression of HP1b causes pericentromeric heterochromatin decompaction accompanied by a reduction in binding of HP1a to H3K9me2, suggesting that the presence of HP1b prevents the function of HP1a in heterochromatin [129]. Moreover, when the N-terminal and hinge regions of HP1α are exchanged into HP1β, chimeric protein droplets are formed [128]. This competition leads to differences in the paralog location, and a gradient is observed in which HP1a or α is found in heterochromatic regions characterized by potent DNA compaction and phase separation activities. This is followed by HP1b or β in areas where there is a change from heterochromatin to euchromatin, whereby the enrichment of HP1b or β works as a bridge, allowing for the recruitment of gene activators which contribute to maintaining open chromatin states. Finally, HP1c or γ are found in euchromatin areas with entirely different partners [71,125,130]. This process requires different interactors, and the different variables may contribute to dissolving the phases [128]. In the following sections, we focus our attention on the described interactions with HP1a to better understand how these interactions regulate heterochromatic domains.

## 3. HP1a Conserved Domains Direct Specific Protein Interactions

In addition to histone recognition, interactions with non-histone chromosomal proteins might serve as an additional mechanism for association between HP1a and chromatin. In Table 1, we detail the direct HP1 interactors revealed in humans, mice, and flies using different methods, such as yeast two-hybrid and pull-down assays. In some cases, the interactions have been confirmed through other indirect methods, such as IP, WB, and IF. In 2014, Alekseyenko et al. performed BioTAP-labeling of the HP1a protein and described new HP1a-binding proteins in addition to RNAs [68]. To characterize the organization and regulation of heterochromatin, Swenson et al. isolated some previously known HP1a interactors as well as others that were completely new. The authors also showed the distribution and dynamic localization patterns during the cell cycle of some interaction partners [70]. In Table 2, we list the HP1a interactors in *Drosophila melanogaster*—not necessarily direct interactors—for which the exact domain(s) of interaction within HP1a have not been characterized.

To identify putative direct HP1a interactors among those presented in Table 2, we searched for the PxVxL motif, a pentapeptide known to bind between the CSD dimer interface formed through the C termini of HP1 [33,47,166,167,168]. The following proteins contain this motif (Figure 3): Dp1, CG15356, Eyg, Woc, Su (var) 2–10, STAT92E, MED26, Vtd, dADD1, CG43736, DNApol-ɛ255, Gnf1, and Sov.

Other motifs can also bind to the HP1 CSD domain. These are known as degenerate motifs, which retain similar characteristics to the classical PxVxL pentamer; some of them have conserved V and L residues, such as LxVxL and CxVxL [96,168,169]. The following proteins were found to contain the LxVxL motif: Arp6, ACF, Qin, Fru, Su (var) 2–10, Hmr, STAT92E, MED26, HIPP1, Odj, Tea, CG43736, E (var) 3-9, CG1815, Rrp6, Rictor, and Gnf1.

We also searched for and found the degenerate motif CxVxL [76] in the following proteins: Su(var) 3-3, POF, ACF, mu2, Bon, Fru, Woc, Tea, CG1815, Rictor, and Sov.

Furthermore, some proteins harbor more than one motif (represented as a yellow box; Figure 3), including WOC (without children), which encodes a transcription factor with zinc fingers and an AT-hook domain for sequence-specific DNA binding. WOC is involved in telomere capping and transcriptional regulation, and was also found to be co-immunoprecipitated and show immunofluorescent co-localization with HP1b and c [75,125]. Additionally, the signal transducer and activator of the transcription protein at 92E (Stat92E), which encodes a transcription factor that shuttles between the cytosol and nucleus and functions in the JAK/STAT pathway [153], presented one PxVxL motif and one LxVxL motif. These motifs can accommodate several interaction options—for example, an interaction with both HP1a and HP1b or c, or even an interaction with other proteins.

Gene ontology analysis revealed that all proteins from Figure 3 are chromatin proteins and that no other processes were enriched (data not shown). Moreover, 69% of proteins (9 of 13) from Table 1, with known direct binding to CSD, were found to have the PxVxL or similar motifs. However, only 35% of proteins (30 of 86) presented in Table 2 were found to have the PxVxL or similar motifs. This analysis indicates that many proteins from Table 2 could interact with HP1a indirectly (i.e., by association with other direct interactors or via RNAs [68,113]. Another possibility is that proteins from Table 2 lacking PxVxL or similar motifs may interact with HP1a directly, but via unknown motifs. In addition, some proteins appear to have more than one possible motif for binding, which may give more weight to the theory that they can bind in different complexes—i.e., if one site is occupied, the other can be used, depending on the partners with which they are associated, thus representing another level of regulation.

Although many of the proteins described here harbor the HP1a-binding domain (Figure 3), this domain is not necessarily the only feature required to establish an interaction with HP1a; binding with other residues may be required to form a stable interaction. Therefore, it will be important to analyze the +1 and −1 positions of the PxVxL motif, where V is position 0. These amino acids could also intervene in different ways to facilitate protein binding. Furthermore, it will be essential to identify whether different degenerate motifs have different affinities, i.e., CxVxL or LxVxL, since we found several proteins that contain more than one domain for binding with HP1a.

## 4. HP1a Interaction with Insulator and Architectural Proteins

Insulators were first defined as regulatory elements that maintain the correct separation of different gene domains, thereby preventing enhancer–promoter communication and/or blocking the expansion of heterochromatin silencing. They also mediate intra- and interchromosomal interactions, which are involved in the large-scale organization of the genome [170,171].

Since the 1980s, it has been known that there are DNA sequences that delimit and isolate a region of chromatin in the *Drosophila* heat shock locus [172,173]. Since then, many of these sequences and the factors that bind to them have been characterized [174,175]. CTCF (CCCTC binding factor) was initially identified as a repressor capable of binding to promoters in chicken and mammalian MYC genes [176]. CTCF was later shown to have an insulator function because it indirectly regulates gene expression by preventing binding between promoters and enhancers or nearby silencers, thus avoiding the inappropriate activation or silencing of certain genes [177]. Recent advances in Hi-C technique have shown that CTCF can mediate the interactions between the boundaries of topologically associating domains (TADs) resulting in the formation of chromatin loops [178]. It should be noted that not all TADs are flanked by CTCF [179].

Although it is not known exactly how CTCF assists in loop formation, a “loop extrusion” model has been proposed. This model suggests that cohesin, which is composed of SMC proteins (structural maintenance of chromosomes) and Rad21 (which is an ortholog of *Drosophila* verthandi (*vtd*)) are directed to the chromatin with the help of the NIPBL protein. Together they “pull” the DNA strand until the cohesin ring is blocked with CTCF [180]. It is currently unclear whether the same mechanism operates in *Drosophila*. However, ChIP-seq experiments have identified several architectural proteins (APs) which are co-localized with CTCF at several sites in the genome.

APs are characterized by promoting contacts between regulatory elements through the formation of loops; thus, they have a role in determining the organization and architecture of chromatin [181,182]. Furthermore, it has been shown that APs can contribute to the establishment of TADs [183]. Some of them include suppressor of hairy-wing (Su(Hw)) [184], dCTCF, the *D. melanogaster* ortholog of mammalian CCCTC-binding factor [185], Boundary Element Associated Factors (BEAF-32A and B) [183], GAGA Associated Factor (GAF) [186], and Zeste-white 5 (Zw5) [187]. Others that have recently been identified include Elba (made up of 3 proteins), Elba1, Elba2, and Elba3 [188], Pita, the zinc-finger protein interacting with CP190 (ZIPIC) [181], Clamp [189], and Ibf-1 and 2 [190].

Typically, the proteins bound to the insulator sequences are necessary but not sufficient for the activity of insulators. Several cofactors are also required to establish physical contacts and anchor them to nuclear structures. In *Drosophila*, proteins such as the modifier of mdg4 (Mod (mdg4)) [191,192] and centrosomal protein of 190kDa (CP190) [193]; cohesins (like Rad21/Vtd); and condensins (like Cap-H2) are present in different combinations in all types of insulators and fulfill these functions [194].

A protein which interacts with HP1a and has a possible role in insulator function is HIPP1 (HP1 and insulator partner protein 1) [68]. Our bioinformatic analysis has shown that HIPP1 could directly associate with HP1a via a PxVxL-related motif (Figure 3b). This protein has a crotonase-fold domain, which makes it a homolog of the human protein chromodomain Y-like (CDYL). In vertebrates, this protein is involved in negatively regulating crotonylation, a modification associated with active promoters [195,196,197]. The function of this protein during development is not essential [198]. Its location is mainly pericentric, but it also binds to several euchromatin regions and interacts with AP proteins such as Su(Hw), Mod(mdg4), and CP190 [199]. HIPP1 functions stabilize the interactions between CP190 and the Su(Hw)-dependent complex [200]. In this article, we include HIPP1 as a possible architectural protein in Table 3, although this function has not yet been fully demonstrated. We found that the degenerate motif of binding with HP1a (Figure 3b) is located at the most terminal part of the protein. The degenerate motif begins at amino acid 752, and the crotonase domain extends from amino acid 675 to 778. This suggests that HIPP1 interactions with HP1a could affect (negatively or positively) its crotonase activity to some extent. Moreover, the crotonase domain seems to have a role in the interaction of HIPP1 with Su (Hw). Presently, there are no data on whether crotonase activity could be affected by interactions with these proteins. This is an important question to address experimentally in the future.

Since HIPP1 contains possible motifs for direct interactions with HP1a, we examined whether such motifs are also present in other APs. The results are summarized in Table 3. The cohesin complex is an important factor in maintaining the structure of chromosomes. In mammals, the cohesin complex co-localizes with CTCF throughout the genome. In many of these sites, CTCF performs its function of enhancer-blocking [201,202]. In *Drosophila,* there is no co-localization observed between CTCF and the cohesin complex [203], but cohesins do co-localize with other APs, such as CP190 [193]. Through a ChIP-chip analysis, it was determined that Nipped-B and cohesins are located preferably in active sites and are absent from the silenced sites [204].

The Rad21 homolog in *Drosophila* is known as Vtd which is a cohesin subunit involved in the ring formation of the cohesion complex. According to our analysis, Vtd has a PxVxL motif responsible for a possible direct binding with HP1a (Figure 3a). HP1a also interacts with the Nip-b protein, which topologically loads the cohesin ring complex onto the chromosomes. The existence of the *Drosophila* ortholog of NIPBL was also confirmed using mass spectrometry, but we could not find a motif in this protein that may be involved in direct binding to HP1a (Table 3).

The Cap-H2 protein has an LxVxL binding motif beginning at 159 aa (N-terminal); also, in Stromalin (SA), this domain is present in the middle of the protein, and the same domain was found in Smc2 in the C-terminal. In Elba3, the canonical binding site, PxVxL, is found at 156 aa and for GAF, this canonical motif is present in the N-terminal. Finally, Zw5 has a putative binding domain, CxVxL, at 449 aa, the most C-terminal part of the protein (Table 3). This opens the door to possible interactions of HP1a with architectural complexes which, in the future, would be interesting to address experimentally.

To better understand whether HP1a is co-localized with APs, we analyzed previously published modENCODE chromatin immunoprecipitation (ChIP-seq) data in *Drosophila* for the region spanning *Abd-A* and *Abd-B* loci (Figure 4). This region has very complicated regulation, and the insulator function is essential for the correct expression of genes within this region [205,206,207] such that the active or silenced state of one domain does not extend to an adjacent one. ChIP assays detected the presence of APs such as CP190 or CTCF, not only at the border elements but also at the promoters of genes such as *Abd-b* [203,208,209]. The interactions between protein insulators, on the one hand, manage to form loops that leave out entire domains. On the other hand, elements in the domain are brought in close contact with the gene promoter, thereby mediating correct gene expression [210,211,212].

As shown in Figure 4, both HP1a (dark red) and HIPP1 (pink) are present at the Fub insulator where APs, such as CP190, Su (Hw), CTCF, and Mod (mdg4) are also observed (the violet box where the Fub insulator region is present). Other known insulators are Mcp and Fab8 (highlighted in the middle and right violet boxes) [213], where APs can be observed but do not co-localize with HP1a. Moreover, not all architectural proteins are present in all the insulator loci at the same time. The *Abd-b* gene is silenced in S2 cells, and HP1a appears enriched at the *Abd-b* promoter and is flanked by APs.

For the architectural proteins with available ChIP-seq data, we analyzed the percentage of peaks that co-localized with HP1a peaks genome-wide and estimated the significance of co-localization using the permutation test at a confidence of 95% (Appendix A). The co-localization between peaks of HP1a and HIPP1reaches 42% (*p*-value 0.0181), followed by Su (Hw) (28%, *p*-value 0.0001), CP190 (26%, *p*-value 0.0001), Mod (mdg4) (24%, *p*-value 0.0001), GAF (16%, *p*-value 0.0001). Although 20% of HP1a peaks co-localize with CTCF peaks, this co-localization was not statistically significant (*p*-value 0.4221), the same case was observed for Zw5 (13%, *p*-value 0.0001, z-score-3.876).

The HP1a CSD domain potentially mediates all the interactions with proteins that were evaluated. This indicates that binding possibly occurs with homodimers or heterodimers of HP1a at specific regions of chromatin. The regions where we assessed the presence of architectural proteins were not constitutive heterochromatin; rather, they represent islands of facultative heterochromatin in the euchromatin. Thus, a disruption of heterochromatin may take place, where HP1a dimers cannot be formed. Subsequently, the binding of HP1a with HMTases could be impaired, which would prevent the spreading of heterochromatinization. Chromatin insulators are essential components of genome architecture across eukaryotes [214,215]. It seems plausible that HIPP1, Vtd, Nip-b, and HP1a cooperate to maintain the insulating complexes and define edges of loops, thereby facilitating the correct separation of heterochromatin and euchromatin.

## 5. HP1a Interaction Partners in Silenced Chromatin

Constitutive heterochromatin represents a substantial fraction of eukaryotic genomes; it plays an important role in the maintenance of genome stability and silencing of repetitive elements. Nonetheless, further studies are needed to fully understand its formation and maintenance throughout development and cell differentiation. Thorough localization studies of HP1a in *Drosophila* and mammals have shown that HP1a proteins associate with regions of constitutive heterochromatin around the centromeres and at the telomeres which are rich in repetitive DNA sequences. For example, in polytene chromosomes of *Drosophila*, mainly the chromocenter (i.e., regions of pericentric chromatin) and the telomeres are stained with HP1 antibodies [22].

Constitutive heterochromatin is established early in development. In *Drosophila,* it starts during MBT (in cycle 13) [216,217]. The proposed model involves a complex that contains a methyltransferase (Eggless/SetDB1) of histone H3 lysine 9 and HP1a. The histone mark H3K9me (di or tri methylated) acts as a binding site for HP1a, which binds through its CHD to these chromatin marks, possibly with the involvement of other stabilizing interactions [51,56,57]. It is known that HP1a crosslinks nucleosomes which form condensed heterochromatic structures. For example, in yeast, HP1a also strengthens the association of the HMTase SUV39H1 to chromatin [49]. SUV39H1 methylates nearby unmethylated H3 tails at lysine 9 via its SET domain, creating new H3K9me-binding sites for HP1a. Thus, this three-component system could explain the spreading and maintenance of heterochromatic gene silencing [218,219]. The structures may then be further stabilized through interactions with additional heterochromatic factors. This interaction is preserved in both mammals and *Drosophila* (see Table 1) [49,72,73].

Several groups have carried out chromosomal rearrangement experiments where a euchromatic gene was translocated to a heterochromatic environment and, as a result of being present in this environment, became silenced with the help of several factors, mainly HP1a [38,220,221]. Subsequently, experiments were carried out to direct HP1a to euchromatin regions, such as region 31 of the *Drosophila* 2L arm. Three of the four studied genes within this region were silenced by HP1a and the methyltransferase Su(var)3-9 [222]. These studies demonstrated HP1a to be an essential protein that promotes heterochromatin formation and gene silencing.

Different methyltransferases can work in conjunction with HP1a. For example, in null HP1 mutants, localization of Su(var)3-9 is no longer limited to the chromocenter but spreads across the chromosomes [72]. Studies using mutants suggest that there is a sequential order in which interactions are established [223]. Another member of this complex is the zinc-finger protein Su (var) 3-7, which appears to function as an effector downstream of Su (var) 3-9 and HP1a. This protein has heterochromatic localization, very similar to that of HP1a on polytene chromosomes in pericentric regions and on chromosome 4 [64]. In addition to decreasing the dose of this protein, it reduces PEV [224]. Increasing the quantity of the product of Su (var) 3-7 prompts heterochromatin extension and epigenetic gene silencing [225]. The formation of heterochromatin is a critical developmental process. Su (var) 3-3, whose homolog in mammals is LSD1, removes H3K4me1/2 marks in early embryonic development. This led to the establishment of a balance between demethylase and methyltransferase Su (var) 3-9, contributing to the maintenance of heterochromatic domains [134].

The heterochromatin–euchromatin borders have previously been described cytologically [226,227] and, later, with ChIP-array analysis of genome distribution of H3K9me2 mark, and were named the “epigenomic borders” [16]. Interestingly, the epigenomic borders varied in different cell lines or tissues studied which lead authors to propose that additional mechanisms besides sequence-specific binding can establish these borders [16]. To identify the borders of pericentric heterochromatin domains more precisely, we analyzed publicly available ChIP-seq profiles in S2 cells for HP1a, along with Su (var) 3-9, Su (var) 3-7, and H3K9me3 (Figure 5, see also Material and Methods). We examined a section near the chromocenter (the black circle at the top of thde schematic representation). A clear enrichment of HP1a along with the other examined proteins and histone marks is seen in the pericentromeric region highlighted with a red rectangle. Further from the centromere (7.3 Mb), this enrichment sharply declines thus indicating the border between heterochromatin and euchromatin. APs such as CTCF and CP190 are clearly enriched just after the border in the euchromatin which is consistent with the function of these proteins to keep chromatin domains isolated from each other [170,181,208,209,228,229] and of CP190 to mark active promoters in *Drosophila* [193]. Therefore, APs may play a role in defining this border. Thus, HP1a can cooperate with other factors at these epigenomic borders to maintain a correct chromatin structure. Interestingly, H3K9 acetylation is still present within the beginning of constitutive heterochromatin, co-localizing with CP190 and some CTCF peaks. These bivalent signatures may facilitate pericentromeric gene transcription, as was observed for some genes [230]. Throughout the chromosome, other HP1a sites co-localize with H3K9ac (blue shaded box, Figure 5). The HP1a enrichment sets the epigenomic border for 2R chromosome arm at 7.3 Mb position, while the epigenomic border described in [16] was set at 7.4 Mb (highlighted red square in Figure 5).

Another interactor of HP1a is dADD1 [68]. dADD1 is an ortholog of the N-terminal domain of mammalian ATRX protein [78] and has a motif for interaction with HP1a in the most N-terminal portion (54 aa) (Figure 3a). This is also found within a region conserved between the three isoforms encoded by this gene. We observed that the dADD1-a isoform is the only isoform that immunoprecipitated together with HP1a [78]. Although the other isoforms have the ability to associate and therefore be immunoprecipitated, this was not observed for any of the conditions we have evaluated so far. Rescue experiments in a null *dadd1* background demonstrated that HP1a was restored to the telomeric region when the rescue was performed with dADD1a [231] but not with dADD1b. However, we also observed that upon dADD1a overexpression, HP1a was lost from the chromocenter in a dADD1a dose-dependent manner. When dADD1 proteins have higher than wild-type levels, the polytene chromosomes become decompacted and lose their banding pattern. The HP1a protein and H3K9me3 mark delocalize and acquire a different distribution within the cell nucleus [232].

These results show that dADD1 proteins are regulators of HP1a, likely maintaining the correct local concentration of HP1a oligomers at certain regions, such as the telomeres and pericentric heterochromatin. The over- and underexpression of dADD1 can disturb the concentration of HP1a and likely affect phase transition, which could lead to chromatin instability and alterations in gene expression [232].

Moreover, it has been demonstrated that human HP1α promotes phase separation of heterochromatin from euchromatin [128], which is also exhibited by the *Drosophila* ortholog [128,233]. The HP1 proteins possibly involved in orchestrating these separations and play an important role in defining their possible environments and interactions. For example, HP1a could be enriched at heterochromatin regions together with a methyltransferase (as with dADD1), forming gel-type droplets with a specific environment. Interestingly, dADD1a possesses intrinsically disordered regions in the C-terminal region (650–696, 716–763, 791–1069, 1112–1132, and 1154–1199 aa) that can contribute to phase separation [234].

The contribution of HP1a to the maintenance of telomeric heterochromatin works via two main functions: as part of the CAP along with proteins such as HOAP (cav) [106,107] and the repression of telomeric retrotransposons in cooperation with piRNAs [235]. Notably, HP1a localization at this heterochromatic site does not depend on its chromodomain but rather on its interaction with dADD1a in somatic cells [231]. The interaction of HP1a and dADD1 at the telomeric region seems to be conserved because ATRX also co-localizes with HP1α at the telomeres in human cells. HP1a’s functions at telomeres seem to depend on the interactions of different proteins and even RNAs. In mammalian embryonic stem cells, ATRX (Alpha-Thalassemia with mental Retardation X-related) [236] has been shown to complex with TRIM28 and SETDB1 recruited by H3K9me3 in retrotransposon regions [237,238]. We have described the interaction of the helicase part of ATRX (XNP) together with the ADD domain (dADD1) in *Drosophila* in conjunction with HP1a. Furthermore, Kuroda’s laboratory was able to immunoprecipitate EGG/dSETDB1 and the Bonus protein (Trim28 homolog) together with dADD1 [68]. These proteins could form a complex and perform a similar function to their mammalian counterparts at retrotransposon regions in *Drosophila*.

## 6. HP1a Interaction Partners in Euchromatin

It is well known that HP1a does not only localize to regions of constitutive heterochromatin. A fraction of HP1a is also present within euchromatic regions of the chromosomes. For example, in polytene chromosomes of *Drosophila*, HP1a is found at about 200 sites within the chromosome arms [14,127]. Moreover, using DamID technique numerous HP1a binding regions within euchromatic parts of *Drosophila* chromosome arms from various non-polytene tissues were revealed [15,17,239,240]. This points to the repressive action of HP1 within euchromatin, an interpretation supported by studies demonstrating the recruitment of HP1a by different transcriptional repressors [81,82] and the reported upregulation of some HP1-bound genes in euchromatin upon mutation of HP1a in *Drosophila* [222].

Numerous experimental data demonstrate that HP1 by itself can induce heterochromatic structures and may, in fact, directly stimulate gene silencing within euchromatin. When HP1 binds to the euchromatin regions of *Drosophila* chromosomes through an ectopic binding domain, this process is almost always sufficient to enable the establishment of heterochromatin and silence neighboring reporter genes [241]. Furthermore, redirecting HP1α or HP1a through a GAL/lacR system to euchromatic regions in mammalian or *Drosophila* cells causes local condensation of higher-order chromatin structure and gene repression [3]. These experiments suggest that HP1 could indeed play a role in gene repression within euchromatic regions of chromosomes. The participation of HP1 in the regulation of euchromatic regions is even more complex and goes beyond its well-established role in gene silencing.

Unexpectedly, at some euchromatic loci, HP1 association clearly corresponds to the elevated gene expression. For example, HP1a is recruited to some of the ecdysone-induced or heat shock-induced puffs of *Drosophila* polytene chromosomes, generated due to strong decondensation of chromatin linked to gene activation [242], implying the modulating role of HP1a in their expression. The association of HP1 with such sites seems to be RNA-dependent since HP1 association with Hsp70 heat shock locus is only observed in the presence of RNA [242]. We also observed dADD1 enrichment at the Hsp70 locus under native conditions [232] (Figure 6). When heat shock is induced, this area becomes free of nucleosomes, the poised RNA pol II begins to elongate which results in rapid activation of transcription [243]. Another group also analyzed the presence of Xnp (the helicase part that completes the mammalian ortholog ATRX) at this locus along with Hira and the H3.3 histone variant. The authors proposed that Xnp could recognize nucleosome-free sites and work to avoid transcriptional problems or defects in subsequent DNA repair [244]. Many questions remain to be answered regarding the role of these proteins at this locus. For example, whether they strengthen the ability of HP1a to maintain a silenced state or if their presence is necessary to promote rapid transcription upon cell insult.

Furthermore, some genes located within pericentric heterochromatin require a heterochromatic environment for their normal expression. The mutations of HP1a reduce the expression of *light* and *rolled* genes, the first to be described [245]. These genes are essential for the organism and reside naturally in heterochromatin blocks on chromosome 2. Later studies observed that a complex of HP1a and Su(var)3-9 generates very compact chromatin in these blocks [246].

High-resolution mapping of the HP1a-binding sites in *Drosophila* shows that HP1 is excluded from the promoter and is restricted to the transcribed regions of actively expressed genes [135,239]. Furthermore, HP1a depletion causes downregulation of a subset of active genes [247]. The interaction of HP1 with RNA may mediate the association of HP1 at euchromatic regions within the genome. HP1a interactions with RNA, most likely in collaboration with other interactions, recruit HP1a to these sites, where it plays a role in the promotion of gene expression [7]. This is supported by studies indicating that the section of HP1a responsible for RNA binding is the hinge [26].

It has been observed that HP1b and HP1c isoforms are more localized in the euchromatic chromosome arms than HP1a [47,222,248,249]. Lee et al. have shown that all HP1 isoforms interact with the phosphorylated at serine 2 or 5 RNA pol II, but do so with different affinities [53]. Various localization patterns of HP1 isoforms on chromosomes may be mediated by different complexes in which these isoforms are involved (Figure 7).

## 7. Future Directions

Although there have been detailed genetic and biochemical studies of HP1′s roles in heterochromatin establishment and maintenance, its position at euchromatic regions and in association with RNA has not been thoroughly characterized. Valuable studies have shed light on the multiple proteins that interact with HP1a. In this extended review, we addressed whether the proteins found in association with HP1a could bear putative motifs that allow direct interaction with HP1 proteins. The identification of HP1a interactors at different chromatin regions is essential to understand the different roles of these protein complexes. We found that among the reported interactors, only a handful conserve motifs for CSD domain interactions (Figure 3). Experimental studies to test if these motifs function in vivo in binding to CSD would be of great importance and could extend our understanding of the biological significance of the interactions.

We also raise the possibility of HP1 interactions with architectural proteins. Indeed, we found that several architectural proteins harbor conserved putative HP1-interacting motifs (Table 3). Further experimentation will be required to understand the role of HP1a in conjunction with architectural proteins and their possible cooperation in the organization of chromatin structure [71].

Very recently, it has been shown that pericentric heterochromatin also establishes well-defined territories through contact with different proteins via the H3K9me2 mark. Most importantly, the maintenance of these territories and the established pericentromeric contacts also influence active genome regions. The protein complexes associated with these domains could also have an essential role in the formation of higher-order chromatin structures [250].

## 8. Materials and Methods

Protein domain structure illustration

The search for possible motifs for interaction with HP1a in protein sequences and the representation of the location of the domain were executed using Python 2.7. The obtained data are presented in Figure 3 and Table 3.

Alignments and phylogenetic inference analyses

Multiple protein sequence alignments were performed using MUSCLE [251]. Maximum likelihood phylogenetic analysis was computed by PhyML [252] using a Dayhoff matrix as a substitution model with 100 bootstraps. The tree was edited using FigTree with the protein domain architecture information of chromodomain and chromoshadow of the containing proteins as predicted by ScanProsite [253]. The data are presented in Figure 2.

ChIP-seq analyses and data visualization

All datasets used in this study were retrieved as processed data from the GEO Omnibus database (http://www.ncbi.nlm.nih.gov/geo/) and shown in Supplementary Material Appendix A [254,255]. ChIP-seq data were visualized using the Integrative Genome Browser [256]. Values from all ChIP-seq datasets represent log2 enrichment values, except in the case of Pol II and H3K27ac, where the values are the combined counts of fragment centers for all replicates.

The percentage of co-localizing peaks between HP1a and the architectural proteins was obtained with the R Bioconductor package. To select the epigenomic borders, the global height of HP1a ChIP-seq peaks summit was measured along the genome. The median height was obtained, this provided a value of 0.9 arbitrary units (a.u.). Chromosome 2R peaks were analyzed every 0.5 Mb from the centromere to the telomere. The median of HP1a peaks near the chromocenter has a value of 3.12 a.u., and it extends to approximately 7 Mb, where it changes to 1.4 a.u. and then decays to 0.5 a.u. When the value decayed below the global median, the edge was marked. Importantly, Su (var) 3-9 and H3K9me3 peaks presented a similar behavior. Genomic coordinates were converted from dm3 to dm6 using FlyBase [257]. The epigenomic boundary set with HP1a was at 7.3Mb whereas in Riddle et al. [16] it was at 7.4 Mb in chromosome 2R. All the coordinates are from the reference genome dm6. Data obtained with these tools are presented in Figure 4, Figure 5 and Figure 6 and S1.

## Figures and Tables

**Figure 1 cells-09-01866-f001:**
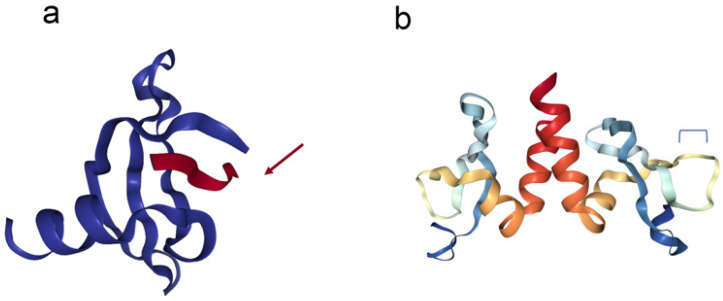
The crystal structure of the CHD (left) and CSD (right) of HP1a. (**a**) The left image is a representation of the chromodomain (blue ribbons) of HP1 complexed with histone H3K9me3 from *Drosophila* (red ribbon mark with red arrow). The CHD (69 aa in length), is made up of three β-sheet antiparallel chains flanked by an α-helix on the C-terminal. The histone tail (16 aa) inserts as a β-strand, completing the β-sandwich architecture of the CHD. (**b**) On the right side is the CSD with the C-terminal region (rainbow ribbons) of HP1a from *Drosophila*. The CSD (87 aa) is a dimeric domain and consists of three antiparallel β-sheet chains flanked by two α-helices. The blue bracket represents the interaction site of the PxVxL peptide. Images were created with the PDB (Protein Data Bank) ID 1KNE [36], 3P7J [42], and NGL Viewer [43].

**Figure 2 cells-09-01866-f002:**
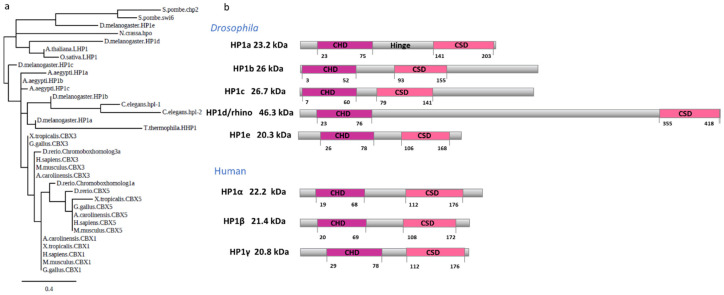
HP1 is preserved during evolution. (**a**) Maximum likelihood phylogenetic analysis of the HP1 protein information of the containing proteins as computed by PhyML. The amino acid sequences were analyzed from the following organisms: *Tetrahymena thermophila*, *Schizosaccharomyces pombe*, *Neurospora crassa*, *Arabidopsis thaliana*, *Oryza sativa*, *Caenorhabditis elegans*, *Aedes aegypti*, *Drosophila melanogaster*, *Xenopus tropicalis*, *Anolis carolinensis*, *Danio rerio*, *Gallus gallus*, *Mus musculus*, and *Homo sapiens*. (**b**) Diagram of HP1 proteins in *Drosophila* and humans. The chromodomain is shown in magenta, and the chromoshadow is in rose. The molecular weight is indicated to the left, with the amino acid localization of the domains displayed below each protein.

**Figure 3 cells-09-01866-f003:**
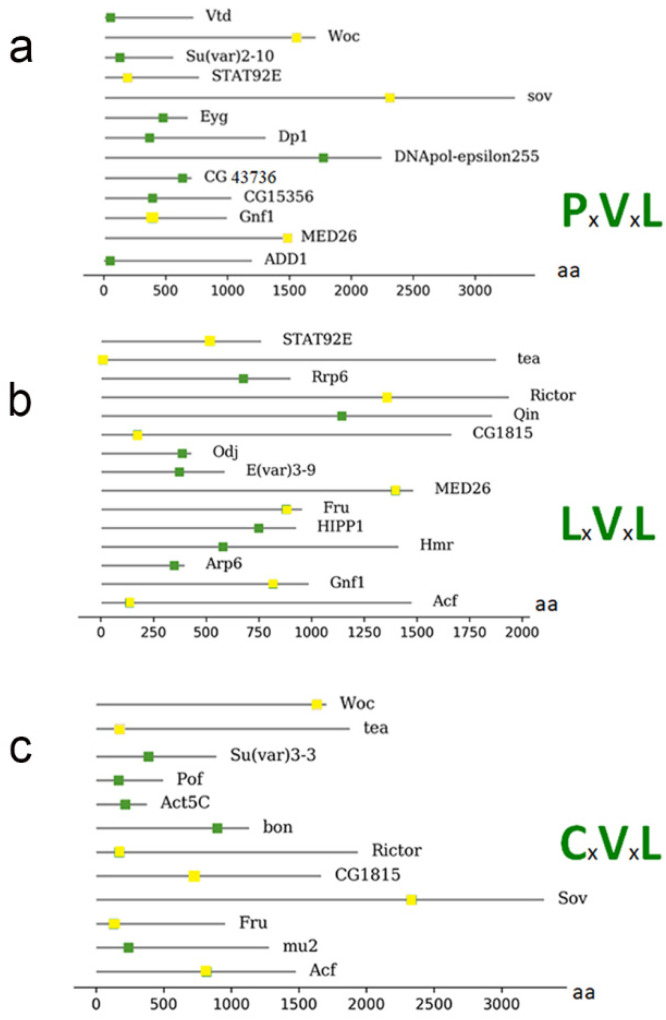
Representation of proteins that have a possible motif for interaction with HP1a from Table 2. (**a**) The proteins connected to the motif PxVxL and the location of the motif within the amino acid sequence. (**b**) Illustration of the proteins with the LxVxL motif and the location of the motif within the amino acid sequence. (**c**) Illustration of the proteins with the CxVxL motif and the location of the motif within the amino acid sequence. The bottom bar indicates the position of the amino acids within the proteins. Proteins that present more than one motif are repeated, and the motif is represented as a yellow box.

**Figure 4 cells-09-01866-f004:**
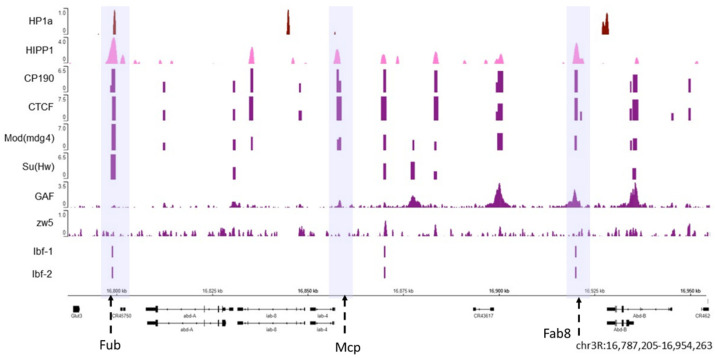
The HP1a and HIPP1 proteins co-localize at homeotic genes along with the AP. The HP1a and HIPP1 proteins co-localize at the Fub insulator (the first violet box) in the *Abd-a* gene along with CP190, Su (Hw), CTCF, and Mod (mdg4), and Ibf 1 and 2. From the previously published ChIP data on S2 cells, we find HP1a (dark red), HIPP1 (pink), and some architecture proteins (purple). The adjacent insulators Mcp and Fab8 do not co-localize with HP1a (center and right violet boxes). The regions with insulators are marked with dotted arrows inside a violet shadow. At the bottom are the *Abd-a* and *Abd-b* genes and their locations; the reference in kilobases.

**Figure 5 cells-09-01866-f005:**
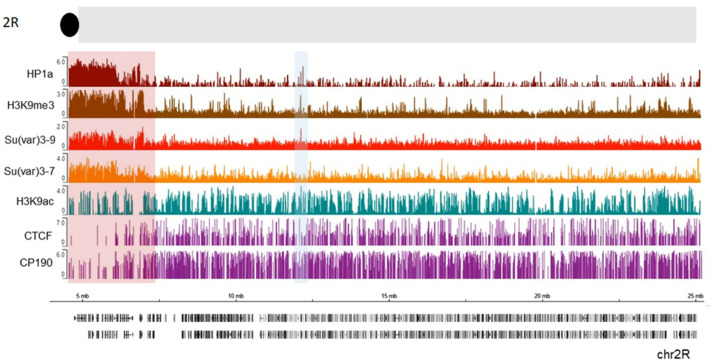
HP1a, together with other proteins, delimits epigenomic borders. Previously published ChIP data on S2 cells were used, where we see HP1a (dark red), Su(var) 3-9 (light red), Su(var) 3-7 (orange), H3k9me3 mark (brown), H3k9ac (green), CTCF, and CP190 (purple). The regions with a pericentric border are marked with a red rectangle according to Riddle et al. The co-localization of HP1a with H3K9ac mark is shaded in blue.

**Figure 6 cells-09-01866-f006:**
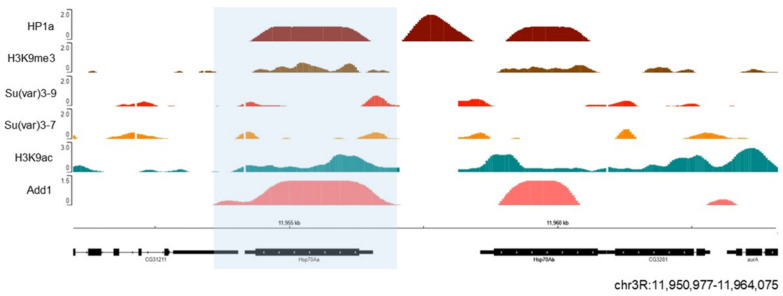
HP1a and dADD1 proteins co-localize at the Hsp70 locus. Previously published ChIP-seq data on S2 cells were used, where we see HP1a (dark red), dADD1 (salmon) HP1a, Su (var) 3-9 (light red), Su (var) 3-7 (orange), H3k9me3 mark (brown), and H3k9ac (green). At the bottom are the gene Hsp70 Aa (blue box) and its location, with the reference in kilobases.

**Figure 7 cells-09-01866-f007:**
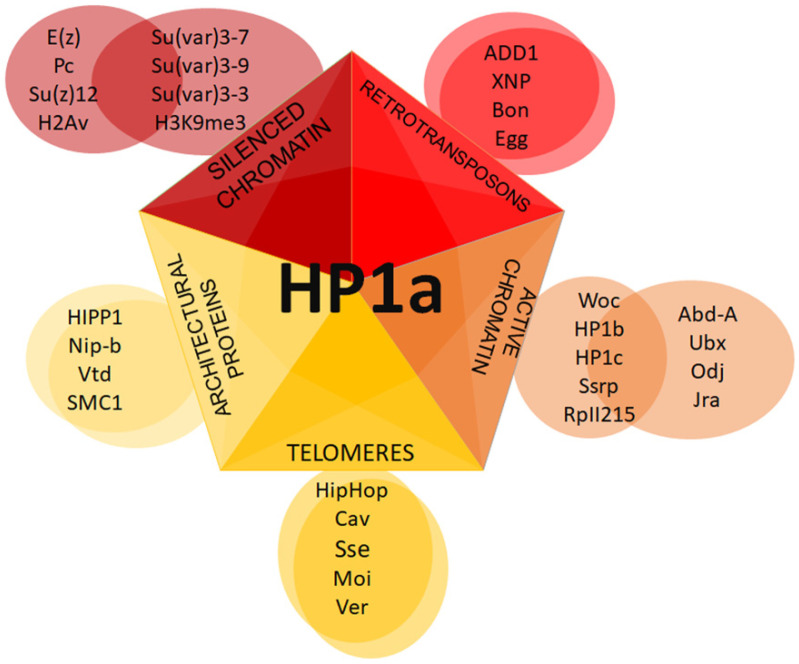
Putative HP1a complexes at different chromatin regions. Schematic representation of distinct complexes formed with HP1a.

**Table 1 cells-09-01866-t001:** Proteins or factors in mammals and *Drosophila* that were revealed as directly bound with the known domains of HP1a/HP1α.

Protein or Cellular Component	Organism	Methodology	References	
**CHD**				
Methylated H3K9	*Drosophila*	IF, FAITC, NMR	[51]	
H2Av	*Drosophila*	IF, tagIP, rPD,	[52]	
RpII215	*Drosophila*	IP, WB, rPD	[7,53]	
Nuclear envelope	Mouse	IF, BA	[54]	
H3	Mouse	IP, FW, rPD	[31,55]	
H1	Mouse	rPD, FW	[55]	
Methylated H3K9	Mouse	rPD	[56]	
Methylated H3K9	Human	rPD, SPRA	[57]	
CTIP2	Human	rPD, IP	[58,59]	
Methylated H1.4K26	Human	BD, IP, rPD, IF	[31,37]	
DNMT1	Human	rPD	[60]	
**CSD**				
Hip/HP4	*Drosophila*	Y2H, tagIP, rPD, IF, tag-WB	[61,62]	
AF10/Alh	*Drosophila*	transPD	[42,63]	
Su(var)3-7	*Drosophila*	Y2H, IP, IF, WB	[64,65]	
PIWI	*Drosophila*	Y2H, IP, IF, NMR, Y2H	[42,66,67]	
Kdm4A	*Drosophila*	transIP, tag-WB, MW, WB, fingerprinting, MS	[68,69,70]	
Ssrp	*Drosophila*	transPD, WB, IP, tandem affinity technology, tagIP, tag-WB	[53,71]	
Su(var)3-9	*Drosophila*	transPD, At, ATC, WB, Y2H, IP, tag-WB, tagIP, fingerprinting, IF, MS	[53,68,70,72,73]	
Su(var)2-HP2	*Drosophila*	IP, At, Y2H, NMR, FAITC, PP, tagIP, fingerprinting, co-sedimentation, molecular weight, molecular sieving, MS	[42,68,70,74,75]	
XNP/dATRX	*Drosophila*	transIP, transPD, MS, IF, WB	[68,76,77,78]	
HP6	*Drosophila*	IP, WB, transPD, tag-WB	[79]	
egg	*Drosophila*	transIP, fingerprinting, rPD	[53,68]	
G9a	*Drosophila*	IP, WB, rPD	[53]	
ova	*Drosophila*	IP, Y2H	[80]	
HP1-BP84	Mouse	Y2H	[81]	
TIF1α	Mouse	Y2H, rPD	[50,81,82]	
CAF-1 p150	Mouse	Y2H, rPD, IF, GFC, NMR	[33,83]	
mSNF2β	Mouse	Y2H	[50]	
KAP1/TIFβ	Mouse	IP, rPD, IF, SPRA, GFC	[50,55,83]	
H4	Mouse	In vitro cross-linking	[31]	
MeCP2	Mouse	tagIP	[84]	
KAP1/TIFβ	Human	Y2H, IP, rPD, IF, GFC	[83,85]	
SP100	Human	Y2H, rPD, transPD, IF	[86]	
Polycomb	Human	IP, rPD, IF	[33]	
ATRX	Human	Y2H, IF, rPD	[85,87]	
CAF-1 p150	Human	rPD	[50]	
Ku70	Human	Y2H, IP, rPD	[88]	
TAF_II_130	Human	Y2H, exPD, transPD	[89]	
Ki-67	Human	Y2H, exPD, IF, ChIP	[90]	
BRG1	Human	IP, rPD, TransPD, IF	[91]	
SUV39H1	Human	rPD, Y2H	[49]	
NIPBL/hScc2	Human	rPD	[85,92]	
HP1-BP74	Human	rPD	[85]	
LBR (Lamin B receptor)	Human	rPD, Y2H, IP	[85,93]	
Sgo1	Human	Y2H, MS. IP	[92,94]	
POGZ	Human	Y2H, MS	[92]	
BARD1	Human	tragIP, transPD	[95]	
KDM2A	Human	IP, transPD, IF	[96]	
LRIF1	Human	IP, transPD	[97]	
Haspin	Human	tragIP, rPD	[98]	
MacroH2A1.2	Human	IP, transPD	[99]	
**Hinge**				
HP-BP74 H1-like	Mouse	Y2H, FW, rPD	[55,81]	
MITR, HDAC4/5	Mouse	IP, rPD	[100]	
**Combination of Domains**				
ORC1-6	*Drosophila*	tagIP	CHD, CSD	[30]
Mcm10	*Drosophila*	proximity ligation assay, IF, IP, WB, transPD, tag-WB, Y2H	CHD, CSD	[101,102]
SuUR	*Drosophila*	Y2H, rPD, transPD, WB, IP, MS, tagIP, fingerprinting	Hin + CSD	[68,93]
Caf1-180	*Drosophila*	transPD, tag-WB, WB, transIP, fingerprinting, IP	Hin + CSD	[68,103]
Cav/HOAP	*Drosophila*	tagIP, IF, IP, exPD	Hin + CSD	[79,104,105,106,107]
Parp-2	Mouse	rPD	Hin + CSD	[108]
TIf1β	Mouse	rPD	Hin + CSD	[108]
ARFL5	Human	Y2H, rPD	CHD + CSD	[109]
INCENP	Human	Y2H, tranPD	Hin + CSD	[94,110]

Methodology: BA, binding assays; ChIP, chromatin immunoprecipitation; IP, co-immunoprecipitation using extract; exPD, pull-down assay using extracts; FAITC, fluorescence anisotropy, isothermal titration calorimetry; FW, far-western analysis; IF, immunofluorescence co-localization; rPD, pull-down assay using recombinant proteins; tragIP, immunoprecipitation with in-vitro translated protein; transPD, pull-down assay using in-vitro translated protein; SPRA, surface plasmon resonance analysis; Y2H, yeast two-hybrid assay; WB, western blot; NMR, nuclear magnetic resonance; PP, predetermined participant; tag-WB, western blot assay performed when specific antibodies for the protein of interest are not available; At, autoradiography; fingerprinting, peptide mass fingerprinting; MS, identification by mass spectrometry; BiFC, bimolecular fluorescence complementation.

**Table 2 cells-09-01866-t002:** HP1a interactors in *Drosophila* for which the interaction domains within HP1a have not been identified.

Protein or Cellular Component	Methodology	Reference
Arp6	IF	[131,132]
E(bx)	WB	[133]
Nap1	WB	[133]
Su(var) 3-3	IP, WB, transIP, fingerprinting	[68,134]
POF	IF	[135,136]
Ndc80	transIP, fingerprinting	[137]
HP5	MS, IP, WB,	[68,70,138]
Pep	IP, WB	[7]
moi	transPD, tag-WB	[107]
ACF	transPD	[130]
Dp1	IP, WB,	[7]
vig	IP, WB	[139]
vig2	IP, WB, rPD	[139]
Hmt4-20	IF	[62]
dre4	tandem affinity purification, multidimensional protein identification technology, WB	[71]
ver	transPD, tag-WB	[140]
HP1c	transPD, WB	[53,71]
Atg8a	PA	[138]
CG11474	PA	[138]
Atf-2	IP, WB	[141]
qin	transIP, WB, IP, tag-WB	[142]
mu2	transIP, WB, Y2H, IP, tag-WB, transPD	[143]
CG15356	FAITC, PP	[42]
jnj	transIP, WB	[144]
SMC5	transIP, WB	[144]
Hrb87F	transIP, WB	[7,145]
Hrb98DE	transIP, WB	[145]
bon	IP, WB	[73]
fru	IP, WB	[73]
eyg	IP, WB, transIP	[146]
Hers	cosedimentation, WB, IP	[73]
woc	FAITC, PP	[75]
H1	rPD, WB, tagIP	[147,148]
Su(var)2-10	IP, At, Y2H, NMR, FAITC, PP, transIP, fingerprinting, cosedimentation, molecular weight, molecular sieving	[42,68,74,75]
Lhr	Y2H, transIP, WB, IP, fingerprinting, tag-WB	[68,149,150,151]
Hmr	IP, WB, tag-WB, transIP, fingerprinting	[68,149,150]
STAT92E	IP, rPD, IF, transPD, tag-WB	[152,153]
MED26	IP, WB, ATC	[154]
MED17	IP, WB	[154]
Incenp	transIP, fingerprinting	[68]
borr	transIP, fingerprinting	[68]
HIPP1	transIP, WB, fingerprinting	[68]
CAP	transIP, fingerprinting	[68]
SMC1	transIP, fingerprinting	[68]
Yeti	transPD, tag-WB	[155]
Mau2	transIP, fingerprinting	[68]
Nipped-B	transIP, fingerprinting	[68]
vtd	transIP, fingerprinting	[68]
Odj	transIP, fingerprinting, Y2H, MS	[68,70,156,157]
vers	transIP, fingerprinting	[68]
HP1b	transIP, fingerprinting, rPD	[53,62,68]
dADD1	transIP, tag-WB, fingerprinting, WB, MS	[68,70,78]
tea	transIP, fingerprinting	[68]
sle	transIP, fingerprinting	[68]
CG43736	transIP, fingerprinting	[68]
E(var)3-9	transIP, fingerprinting	[68]
CG1815	transIP, fingerprinting	[68]
NSD	transIP, fingerprinting	[68]
CG7692	transIP, fingerprinting, MS	[68,70]
CG1737	transIP, fingerprinting	[68]
CG30403	transIP, fingerprinting	[68]
Jra	IP, WB, transIP, MS	[158]
Rrp6	coimmunoprecipitation, tag-WB, transIP, WB	[159]
Pc	IP, WB	[160]
Su(z)12	IP, WB	[160]
E(z)	IP, WB	[160]
HipHop	transPD, WB, IP, chromatography technology, molecular sieving, MW	[105,106]
CG8108	transIP, fingerprinting, MS	[68,70]
Sse	transIP, tag-WB, transPD, WB	[161]
Hsc70-3	MS	[70]
βTub56D	MS	[70]
Chd64	MS	[70]
Hsp83	MS	[70]
Act5C	MS	[70]
rictor	transIP, fingerprinting	[162]
Tsr	MS	[70]
dmt	Y2H, transIP, MS	[163]
DNApol-ɛ255	proximity ligation assay, fluorescence microscopy	[101]
Gnf1	IP, WB, proximity ligation assay, fluorescence microscopy	[101]
Ubx	IF, BiFC	[164]
abd-A	IF, BiFC	[164]
sov	transIP, fingerprinting	[68,165]
H3	exPD, FAITC, FW, NMR, PP	[45,51,53,71,125]
bbx	Y2H	[157]
tj	Y2H	[157]

Methodology: IP, co-immunoprecipitation using extract; exPD, pull-down assay using extracts; FAITC, fluorescence anisotropy isothermal titration calorimetry; FW, far-Western analysis; IF, immunofluorescence co-localization; rPD, pull-down assay using recombinant proteins; transPD, pull-down assay using in-vitro translated protein; Y2H, yeast two-hybrid assay; WB, western blot; PA, predictive algorithms; NMR, nuclear magnetic resonance; PP, predetermined participant; GI, genetic interference; tag-WB, western blot assay performed when specific antibodies for the protein of interest are not available; At, autoradiography; fingerprinting, peptide mass fingerprinting; MS, identification by mass spectrometry; BiFC, bimolecular fluorescence complementation.

**Table 3 cells-09-01866-t003:** Architectural proteins of *Drosophila* and motifs for possible interaction with HP1a protein.

Protein	PxVxL	CxVxL	LxVxL
CTCF			
Su(Hw)			
BEAF-32			
pita			
ZIPIC			
Ibf1			
Ibf2			
Mod(mdg4)			
CP190			
Cap-H2			X
Elba1			
Elba2			
Elba3	X		
Shep			
Zw5		X	
Clamp			
GAF	X		
Nip-b			
Vtd	X		
SA			X
Smc1			
Smc2			X
Smc3			
HIPP1			X

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
