# Peer review of "Insights into HP1a-Chromatin Interactions"

_cells, 2020, doi:10.3390/cells9081866_

Round 1
Reviewer 1 Report
After English editing by MDPI, the manuscript of Meyer-Nava et al. became more readable. However, the text marked by yellow and some other pieces of text still remain not well written. I have performed the whole text editing and below presented my suggestions of how to improve text further. Additionaly, I listed below some of my concerns relative the analytical part of the text.
- Lines 79-81: “In 1930, experiments using X-ray treatment of flies determined that genes were induced to translocate from euchromatic regions to the vicinity of pericentric heterochromatin, acquiring a motley pattern of expression [21].” Change to: “In 1930, experiments using X-ray treatment of flies have shown that genes that were translocated from euchromatic regions to the vicinity of pericentric heterochromatin, acquired a motley pattern of expression [21].”
- Lines 91-94: “Moreover, as mentioned above, HP1 proteins are conserved in a variety of organisms, including fission yeast (as Swi6 and Chp2) [24], [25] and also vertebrates such as amphibians (e.g., frog (xHP1α and xHP1γ)) [26], birds (e.g., chicken (HP1α, HP1β, and HP1γ)) [27], and mammals (such as mouse (HP1α, HP1β, and HP1γ)) [28].” Nothing was mentioned above. So, delete: “Moreover, as mentioned above”.
- Lines 202-204: “Flies also have two germ line-specific isoforms, HP1d and rhino, which are expressed in the ovaries and involved in transposon silencing in the germline via piRNA clusters. HP1e is expressed in the testes and is essential for paternal chromosome segregation through embryonic mitosis [122].” HP1d is a synonym of Rhino. So change the sentence to: “Flies also have two germline-specific isoforms, HP1d (Rhino) and HP1e. Rhino is expressed in the ovaries and involved in transposon silencing in the germline via piRNA clusters [Klattenhoff et al. 2009, DOI:https://doi.org/10.1016/j.cell.2009.07.014]. HP1e is expressed in the testes and is essential for paternal chromosome segregation through embryonic mitosis [122].”
- Lines 231-232: “Although the CHDs of all three HP1 proteins are involved in the recognition and binding of trimethylation, they do bind with the same affinity.” It is a mistake. Change to: “Although the CHDs of all three HP1 proteins are involved in the recognition and binding of H3K9me2/3, they do bind with different affinity.”
- Lines 251-264: In Table 1, we detail the direct interactions as tested in humans, mice, and flies using different methods, such as yeast two-hybrid and pull-down. In some cases, the interactions have been confirmed through other indirect methods, such as IP, WB, and IF, as appropriate. In 2014, Alekseyenko et al. performed BioTAP-labeling of the HP1a protein, where they described new HP1a-binding proteins in addition to RNAs [68]. To characterize the organization and regulation of heterochromatin, Swenson et al. purified HP1a interactors and isolated some interactors that had previously been described but purified others that were completely new. The authors also showed the distribution and dynamic localization patterns during the cell cycle of some interaction partners [70]. In Table 2, we describe the interactions tested by different methods in these two articles. We also compiled other interactions—not necessarily direct interactions—with other proteins defined only in Drosophila melanogaster. In Table 1, the direct interactions are presented. In Table 2, we describe interactions in Drosophila detected using indirect methodologies, such as immunofluorescence and immunoprecipitation, for which the exact interaction domain is unknown.” Change to: “In Table 1, we detail the direct HP1 interactors revealed in humans, mice, and flies using different methods, such as yeast two-hybrid and pull-down assays. In some cases, the interactions have been confirmed through other indirect methods, such as IP, WB, and IF. In 2014, Alekseyenko et al. performed BioTAP-labeling of the HP1a protein and described new HP1a-binding proteins in addition to RNAs [68]. To characterize the organization and regulation of heterochromatin, Swenson et al. isolated some previously known HP1a interactors as well as others that were completely new. The authors also showed the distribution and dynamic localization patterns during the cell cycle of some interaction partners [70]. In Table 2, we list the HP1a interactors in Drosophila revealed by indirect methodologies in these two articles. We also compiled HP1a interactions — not necessarily direct interactions — with other proteins defined only in Drosophila melanogaster.”
- Lines 275-277: “To identify putative direct interactors of HP1a, we searched for the PxVxL motif which is found inserted between the CSD dimer interface formed through the C termini of HP1 [33], [47], [165]–[167] from the interactors in Table 2. The following proteins were identified (Figure 3):” Change to: “To identify putative direct HP1a interactors among that presented in Table 2, we searched for the PxVxL motif between the CSD dimer interface formed through the C termini of HP1 [33], [47], [165]–[167]. The following proteins contain this motif (Figure 3):”
- Lines 296-298: “These motifs can accommodate several interaction options — for example, a motif with HP1a and another with HP1b or c, or even with other proteins, in case the interaction site is occupied.” Change to: “These motifs can accommodate several interaction options — for example, an interaction with both HP1a and HP1b or c, or even an interaction with other proteins.”
- Lines 308-309: “Gene ontology analysis of all these proteins revealed that they are all chromatin proteins and that no other processes were enriched (data not shown).” Change to: “Gene ontology analysis revealed that all proteins from Figure 3 are chromatin proteins and that no other processes were enriched (data not shown).”
- Lines 309-315: “Moreover, the proteins from Table 1, with known binding to CSD, were analyzed and 69% (9 of 13) were found to have the PxVxL motif or a variant. From the total of 86 proteins shown in Table 2, 35% of proteins (30 of 86) were found to have the motif as presented in Figure 3. The fact that the percentage seems low compared to that for the proteins already known to bind from direct experiments could be because many interactions in Table 2 may be regulated by a poorly studied region, such as the hinge [100], by other unknown motifs or be mediated by RNA [68], [113].” Change to: “Moreover, 69% proteins (9 of 13) from Table 1, with known direct binding to CSD, were found to have the PxVxL or similar motifs. However, only 35% proteins (30 of 86) presented in Table 2 were found to have the PxVxL or similar motifs. This analysis indicates that many proteins from Table 2 could interact with HP1a indirectly (i.e., by association with other direct interactors or via RNAs [68], [113]). Another possibility is that proteins from Table 2 lacking PxVxL or similar motifs may interact with HP1a directly, but via unknown motifs.”
- I do not understand by what principle authors have listed some proteins in Table 2, but not in Table 1. According to the notification in Table 2, many HP1a interactors were identified in two-hybrid (Y2H) or pull-down (PD) assays and thus represent the direct HP1a interactors. For example, moi, ACF, vig2, ver, HP1c, mu2, etc. Please, clarify or transfer those proteins to the Table 1. Also it will be better to indicate the presence of the PxVxL or similar motifs in proteins listed in Table 2.
- Lines 328-341: “To better understand the role of HP1a in certain genomic regions, and find putative HP1a complexes, we aimed to find previously published studies in which chromatin immunoprecipitation was followed by high-throughput sequencing (ChIP-seq) analysis of Drosophila S2 cells (modENCODE). We then selected several chromosomal proteins and histone modifications which have publicly available data and represent epigenetic marks enriched at different chromosomal regions (for example, H3K9me3 and Su(var)3-9 for heterochromatin, H3K9ac for euchromatin, and several architectural proteins for insulators). Then, we visualized these data on the IGV genome browser (see Methods section). We found that HP1a co-localizes with several proteins and it may possibly have direct interactions with some components of the architectural protein system (Table 3and Figure 4). We also found well-defined edges as described before by Riddle et. al., that mark both pericentric and telomeric heterochromatin (Figure 5). The co-localization data plus the finding that several putative interactors conserve motifs that could potentially interact directly with HP1a allowed us to propose the composition and location of HP1a containing complexes (Figure 7). Below, we present our findings.” I suggest to delete this piece of text because it is unnecessary.
- Lines 355-364: “Recent advances have determined that CTCF can mediate the relationship between nuclear structures since Hi-C has shown that topologically associating domains (TADs) are separated by regions enriched with CTCF binding sites. However, it should be clarified that not all TADs are flanked by CTCF [177]. Interactions between DNA sequences that are distant from each other are the product of chromatin loops, which are relatively stable and mediated by the presence of CTCF. For this reason, CTCF has been recognized as the only essential protein for the formation of these loops in mammals [178]. Although it is not known how CTCF assists in loop formation, a “loop extrusion” model has been proposed.” Change to: “Recent advances in Hi-C technique have shown that CTCF can mediate the interactions between the boundaries of topologically associating domains (TADs) resulting in the formation of chromatin loops [178]. It should be noted that not all TADs are flanked by CTCF [177]. Although it is not known exactly how CTCF assists in loop formation, a “loop extrusion” model has been proposed.”
- Lines 367-369: “This model is yet to be proven in Drosophila. However, ChIP-seq experiments have identified several proteins as being co-localized with CTCF at several sites in the genome; these proteins are known as architectural proteins (APs).” Change to: “It is currently unclear whether the same mechanism operates in Drosophila. However, ChIP-seq experiments have identified several architectural proteins (APs) which are co-localized with CTCF at several sites in the genome.”
- Lines 381-382: “Typically, the proteins binding to the insulator sequences are necessary but not sufficient for activity in a specific chromatin region.” Change to: “Typically, the proteins bound to the insulator sequences are necessary but not sufficient for activity of insulators.”
- Lines 388-391: “A protein which interacts with HP1a and has a possible role in insulator function is HIPP1 (HP1 and insulator partner protein 1), which was found by the Kuroda group [68]. Through bioinformatic analysis, we discovered that HIPP1 could directly associate with HP1a via a degenerate motif (Figure 3b).” Change to: “A protein which interacts with HP1a and has a possible role in insulator function is HIPP1 (HP1 and insulator partner protein 1) [68]. Our bioinformatic analysis has shown that HIPP1 could directly associate with HP1a via a PxVxL-related motif (Figure 3b).”
- Line 406: I suggest to add the following sentence at the beginning of this paragraph: “Since HIPP1 contains possible motifs for direct interactions with the HP1a, we examined whether such motifs are also present in other APs.”
- Lines 414-415: “In our analysis, Vtd has a PxVxL motif for a possible direct binding with HP1a (Figure 3a).” Change to: “According to our analysis, Vtd has a PxVxL motif responsible for a possible direct binding with HP1a (Figure 3a).”
- I suggest to delete text at lines 420-424: “Therefore, HP1a could bind other APs (such as HIPP1) and contribute to insulator complex activity in these regions. Although such APs have not been detected by the methods used to analyze HP1a interactors, we check for binding sites due to the large number of coincidences that have been found to occur between these proteins. We analyzed the insulator and architectural proteins, cohesin, and condensin containing complexes. The results are summarized in Table 3.”
- Lines 437-439: “To better understand the role of HP1a in complex with APs proteins, we searched for previously published chromatin immunoprecipitation (ChIP-seq) analyses using Drosophila in modENCODE for the region spanning Abd-A and Abd-B (Figure 4).” Change to: “To better understand whether HP1a is co-localized with APs, we analyzed previously published modENCODE chromatin immunoprecipitation (ChIP-seq) data in Drosophila for the region spanning Abd-A and Abd-B loci (Figure 4).”
- Lines 447-451: “As shown in Figure 4, both HP1a (dark red) and HIPP1 (pink) are present at the Fub insulator where APs are also observed, such as CP190, Su(Hw), CTCF, and Mod (mdg4) (the violet box where the Fub insulator region is present). Other known insulators are Mcp and Fab8 (highlighted in the middle and right violet boxes) [212], where more APs can be observed but do not necessarily co-localize with HP1a.” Change to: “As shown in Figure 4, both HP1a (dark red) and HIPP1 (pink) are present at the Fub insulator where APs, such as CP190, Su(Hw), CTCF, and Mod (mdg4) are also observed (the violet box where the Fub insulator region is present). Other known insulators are Mcp and Fab8 (highlighted in the middle and right violet boxes) [212], where APs can be observed but do not co-localize with HP1a.”
- Lines 465-474: “Among the architectural proteins with available ChIP-seq data, we analyzed the percentage of peaks that co-localized with HP1a peaks. We observed that HP1a peaks co-localize with some proteins, so we analyzed the data with permutation test at a confidence of 95% (Supplementary Figure 1). HIPP1 co-localizes in 42% of the peaks (P-value 0.0181), followed by Su(Hw) 28% (P-value 0.0001), CP190 (26%, P-value 0.0001), Mod(mdg4) 24% (P-value 0.0001), and GAF (16%) (P-value 0.0001). The proteins that have a possible binding motif are not necessarily those that share the highest number of peaks with HP1a, such as GAF (16%) and Zw5 (13%), with the exception of HIPP1, which has the highest number of peaks in common, and its interaction has been demonstrated by other experimental means. Also, 20% of HP1a peaks co-localize with CTCF, but the permutation test was not statistically significative (P-value 0.4221).” Change to: “For the architectural proteins with available ChIP-seq data, we analyzed the percentage of peaks that co-localized with HP1a peaks genome-wide and estimated the significance of co-localization using permutation test at a confidence of 95% (Supplementary Figure 1). The co-localization between peaks of HP1a and HIPP1reaches 42% (P-value 0.0181), followed by Su(Hw) (28%, P-value 0.0001), CP190 (26%, P-value 0.0001), Mod(mdg4) (24%, P-value 0.0001), GAF (16%, P-value 0.0001) and Zw5 (13%, P value ???). Although 20% of HP1a peaks co-localize with CTCF peaks, this co-localization was not statistically significant (P-value 0.42).”
- Line 479: “instead, they were more of the facultative type and euchromatic.” Change to: “rather, they represent islands of facultative heterochromatin in the euchromatin.”
- Lines 480-481: “Thus, a dilution of heterochromatin may take place, mediated by architectural proteins, where HP1a dimers cannot be formed.” Change to: “Thus, a disruption of heterochromatin may take place, where HP1a dimers cannot be formed.”
- Line 482: “which would prevent heterochromatinization from spreading.” Change to: “which would prevent spreading of heterochromatinization.”
- Lines 483-485: “It seems plausible that HIPP1, Vtd, Nip-b, and HP1a cooperate to maintain the insulating complexes and define edges of loops, thereby facilitating the correct separation of heterochromatin and euchromatin.”
- Lines 519-521: “For example, in null HP1 mutants, Su(var)3-9 is no longer limited at the chromocenter but roughly spreads across the chromosomes [72].” Change to: “For example, in null HP1 mutants, localization of Su(var)3-9 is no longer limited to the chromocenter but spreads across the chromosomes [72].”
- Lines 528-530: “The enzyme Su(var)3-3, whose homolog in mammals is LSD1, removed H3K4me1 and H3K4me2 marks in early in embryonic development. This led to establishing a balance between demethylase and methyltransferase Su(var)3-9, contributing to the maintenance of heterochromatic domains [133].” Change to: “Su(var)3-3, whose homolog in mammals is LSD1, removes H3K4me1/2 marks in early embryonic development. This led to establishment of a balance between demethylase and methyltransferase Su(var)3-9, contributing to the maintenance of heterochromatic domains [133].”
- Lines 532-570: “To identify heterochromatic domains, we searched public ChIP-seq data while using HP1a as a guideline (dark red), along with Su(var)3-9 (light red), Su(var)3-7 (orange), and H3K9me3 mark (brown) (Figure 5). We examined a section near the chromocenter (the black circle at the top of schematic representation), where we can see a clear enrichment of HP1a along with the other examined proteins and histone marks. In the region highlighted with a red rectangle, we present the pericentromeric chromatin featuring an enrichment of HP1a. Here, we can visualize that this enrichment decays as we move further from the centromere (7.3 Mb). Moreover, APs such as CTCF and CP190 were found to co-localize at the border. There is a clear enrichment of these proteins past the heterochromatin border (purple peaks) since the function of these proteins is not as part of heterochromatin, but to keep the domains isolated from each other and prevent them from affecting each other [169], [180], [207], [208], [225], [226]. In addition, CP190 has been found to mark active promoters in Drosophila [192]. Interestingly, H3K9 acetylation is embedded within this heterochromatic region, which co-localizes with CP190 and some CTCF peaks. This bivalent heterochromatin may facilitate pericentromeric gene transcription, as was observed for some genes [227]. Throughout the chromosome, other HP1a sites co-localize with H3K9ac (blue shade, Figure 5). The epigenetic heterochromatin–euchromatin borders have previously been described cytologically [228], [229] and, later, with ChIP-array analysis, and were named as “epigenomic borders”[16]. Epigenomic borders were established with the H3K9me2 mark, interestingly, the epigenomic borders varied according to the cell line or tissue studied which lead them to propose that additional mechanisms besides the DNA sequence can act to establish these borders [16]. The epigenomic borders near the chromocenters or telomeres can also be visualized when examining HP1a, H3K9me3 and Su(var)3-9 ChIP-seq data (dark red, brown, and orange tracks respectively, Figure 5, see Methods section). Analyzing HP1a enrichment sets the epigenomic boundary at 7.3Mb, while the epigenomic border described in [16] was set at 7.4Mb for Chr2R (highlighted red square in Figure 5). Also, the architectural proteins such as CP190 and CTCF seem to play a role in defining these edges (purple tracks, Figure 5), as the enrichment of these proteins is maintained after signals of H3K9me3 and HP1a decay further away from the centromere. Thus, HP1a can cooperate with other factors at these epigenomic borders to maintain a correct chromatin structure.” Change to: “The heterochromatin–euchromatin borders have previously been described cytologically [228], [229] and, later, with ChIP-array analysis of genome distribution of H3K9me2 mark, and were named the “epigenomic borders” [16]. Interestingly, the epigenomic borders varied in different cell lines or tissues studied which lead authors to propose that additional mechanisms besides sequence-specific binding can establish these borders [16]. To identify the borders of pericentric heterochromatin domains more precisely, we analyzed publicly available ChIP-seq profiles in S2 cells for HP1a, along with Su(var)3-9, Su(var)3-7, and H3K9me3 (Figure 5, see also Material and Methods). We examined a section near the chromocenter (the black circle at the top of schematic representation). A clear enrichment of HP1a along with the other examined proteins and histone marks is seen in the pericentromeric region highlighted with a red rectangle. Further from the centromere (7.3 Mb), this enrichment sharply declines thus indicating the border between heterochromatin and euchromatin. APs such as CTCF and CP190 are clearly enriched just after the border in the euchromatin which is consistent with the function of these proteins to keep chromatin domains isolated from each other [169], [180], [207], [208], [225], [226], and of CP190 to mark active promoters in Drosophila [192]. Therefore, APs may play a role in defining this border. Thus, HP1a can cooperate with other factors at these epigenomic borders to maintain a correct chromatin structure. Interestingly, H3K9 acetylation is still presented within the beginning of constitutive heterochromatin, co-localizing with CP190 and some CTCF peaks. This bivalent signatures may facilitate pericentromeric gene transcription, as was observed for some genes [227]. Throughout the chromosome, other HP1a sites co-localize with H3K9ac (blue shaded box, Figure 5). The HP1a enrichment sets the epigenomic boder for 2R chromosome arm at 7.3 Mb position, while the epigenomic border described in [16] was set at 7.4 Mb (highlighted red square in Figure 5).”
- Line 572: “Another interactor of HP1a is a protein of interest to our laboratory, dADD1 [68].” Change to: “Another interactor of HP1a is dADD1 [68].”
- Lines 592-594: “Moreover, it has been demonstrated that human HP1α promotes phase separation in heterochromatin [127], which is an attribute that is also exhibited by the Drosophila ortholog [127], [232].” Change to: “Moreover, it has been demonstrated that human HP1α promotes phase separation of heterochromatin from euchromatin [127], which is also exhibited by the Drosophila ortholog [127], [232].”
- Lines 596-597: “forming a gel-type drop with a specific environment.” Change to: “forming a gel-type droplets with a specific environment.”
- Lines 616-619: “Although numerous proteins share the same function, for example, methyltransferases, different complexes may develop due to the formation of liquid droplets. Thus, in addition to the separation by heterochromatic edges, inhibition may occur due to the formation of heterochromatic drops that could prevent possible complexes that exist within heterochromatin [127].” I do not understand these sentences and thus recommend to delete them.
- Lines 623-624: “Counterintuitively, it is well known that HP1a does not only localize to regions of constitutive heterochromatin.” Delete “counterintuitively”.
- Lines 624-626: “Within euchromatic regions of the chromosomes, a fraction of HP1a is also present. For example, in polytene chromosomes of Drosophila, HP1a is found at about 200 sites within the chromosome arms [14], [126].” Authors may add several references to the works, where distribution of HP1a in different Drosophila non-polytene tissues or cell types were described: “A fraction of HP1a is also present within euchromatic regions of the chromosomes. For example, in polytene chromosomes of Drosophila, HP1a is found at about 200 sites within the chromosome arms [14], [126]. Moreover, using DamID technique numerous HP1a binding regions within euchromatic parts of Drosophila chromosome arms from various non-polytene tissues were revealed [de Wit et al. 2007, doi: 10.1371/journal.pgen.0030038; Marshall and Brand 2017, doi: 10.1038/s41467-017-02385-4; Pindyurin et al. 2018, doi: 10.1186/s13072-018-0235-8; Ilyin et al. 2020, doi: 10.1007/s00412-020-00738-5]”.
- Lines 626-627: “This points to a positive role for HP1 within euchromatin in the repression of individual genes,” Change to: “This points to the repressive action of HP1 within euchromatin,”
- Line 631: “Following this line of thought, numerous experimental interpretations contend that HP1 …” Change to: “Numerous experimental data demonstrate that HP1 …”
- Lines 635-637: “Furthermore, redirecting HP1α through a GAL/lacR system to euchromatic zones in mammalian cells causes the local condensation of the higher-order chromatin structure [239].” Authors may add the corresponding reference and change this sentence to the following: “Furthermore, redirecting HP1α or HP1a through a GAL/lacR system to euchromatic regions in mammalian or Drosophila cells causes local condensation of higher-order chromatin structure and gene repression [239], [Brueckner et al. 2016, DOI: 10.1186/s13072-016-0096-y].”
- Lines 637-638: “These experiments suggest that HP1 could indeed play a part in the gene repression within euchromatic regions of chromosomes.” Change to: “These experiments suggest that HP1 could indeed play a role in gene repression within euchromatic regions of chromosomes.”
- Lines 648-654: “Unexpectedly, at some euchromatic loci, HP1 association clearly corresponds to elevated gene expression, for example, in some developmental (e.g., ecdysone) and heat shock-induced chromosome puffs in Drosophila, which are morphological hallmarks generated by strong decondensation of chromatin due to high gene expression levels. Moreover, as long as gene expression is induced at these regions of extremely active transcription, Drosophila HP1a is recruited to these decondensed zones of the genome [240], suggesting that HP1 has a role in this expression without inducing heterochromatic structures.” Change to: “Unexpectedly, at some euchromatic loci, HP1 association clearly corresponds to the elevated gene expression. For example, HP1a is recruited to some of the ecdysone-induced or heat shock-induced puffs of Drosophila polytene chromosomes, generated due to strong decondensation of chromatin linked to gene activation [240], implying the modulating role of HP1a in their expression.”
- Lines 654-656: “The association of HP1 with such sites seems to be RNA-dependent, as indicated by the observation that HP1 association with heat shock (such as via Hsp70) is only observed in the presence of RNA [240].” Change to: “The association of HP1 with such sites seems to be RNA-dependent, since HP1 association with Hsp70 heat shock locus is only observed in the presence of RNA [240].”
- Lines 657-658: “When heat shock is induced, this area is free of nucleosomes, and polymerase then arrives, resulting in rapid transcription [241].” Change to: “When heat shock is induced, this area becomes free of nucleosomes, after that RNA pol II arrives resulting in rapid activation of transcription [241].”
- Lines 676-677: “Most likely in collaboration with other interactions, mRNA interactions recruit HP1 to these sites, where the protein plays a role in the promotion of gene expression.” I suggest to add reference supporting this statement: “HP1a interactions with RNA, most likely in collaboration with other interactions, recruit HP1a to these sites, where it plays a role in the promotion of gene expression [Piacentini et al. 2009, DOI: 10.1371/journal.pgen.1000670].”
- Lines 680-700: “A repressed or active chromatin state is involved in the regulation of higher-order structures, such as topologically associating domains. For instance, TAD boundaries coincide with active chromatin marks and insulator proteins. Both active and inactive TADs were found, and their spatial segregation was observed [247]. A high number of polytene bands, at least 95%, correlate with uninterrupted TADs, whereas interbands correspond to inter-TADs. These data suggest a stable state of chromosome decondensation. Recently, Ulianov et al. observed that the limits of the TADs correspond with an increase in the RNA pol II signal and the activation mark H3K27ac [248]. Then, using the Hi-C data from Ramirez et al. [249], we aligned these data along with the ChIP-seq data of HP1a and found that some HP1a peaks present the same binding profile as observed for these transcriptionally active chromatin marks (Supplementary Figure 2 blue shadow boxes). However, we also found HP1a enrichment at intra-TADs (see the magnified image in Supplementary Figure 2b (dark red peaks)). We converted the Hi-C data to be able to perform a permutation analysis with the boundaries of TAD vs. HP1a and no significance was found (data not shown). Active transcription and Pol II presence have emerged as important components in defining higher-order chromatin structures such as TADs. Additionally, HP1a mutants display changes in the patterns of gene expression, not only genes become de-repressed, but also many of them show a decrease in their expression levels [240]. However, a permutation test found that the association of HP1a ChIP-seq peaks with Pol II ChIP-seq peaks is not significant (see supplementary Figure 3). Despite the permutation test was not statistically significant, since HP1a mutants affect transcription, further experiments will be needed to address if HP1a is involved in the establishment or maintenance of higher order chromatin structures such as TADs.” In this part of analysis, the authors tested an idea that HP1a participates in the establishment of TAD borders. However, using permutation test they do not find any non-occasional co-localization between HP1a and TAD borders, at least in S2 cells. This co-localization is also not detectable by visual examination of the Supplementary Fig. 3. These findigs indicate that the initial idea was wrong. The argument that “HP1a mutants affect transcription” and thus may be involved in establishment of TAD borders, is invalid, because many other reasons may be responsible for this effect. So, the negative result presented in this paragraph (lines 680-700) together with its illustration on the Supplementary Fig. 3 should be deleted from the text of the manuscript.
- Lines 701-703: “It has frequently been observed that the other isoforms, either HP1b or c, co-localize better in euchromatin areas, but the possible role of HP1a in these regions has not been studied further [47], [221], [250], [251].” Change to: “It has been observed that HP1b and HP1c isoforms are more localized in the euchromatic chromosome arms than HP1a [47], [221], [250], [251].”
- Lines 703-706: “Despite possessing more information on the other isoforms co-localizing in euchromatin, Lee et al. examined the possibility that all of the isoforms are found together with phosphorylated polymerase, either at serine 2 or in 5 and found that all HP1 isoforms are able to bind but do so with different affinities [53].” Change to: “Lee et al. have shown that all HP1 isoforms interact with the phosphorylated at serine 2 or 5 RNA pol II, but do so with different affinities [53].”
- Lines 706-708: “The small differences that exist between these isoforms could provide more information about the complexes that are formed and their role at different chromatin sites (Figure 7).” Change to: “Various localization patterns of HP1 isoforms on chromosomes may be mediated by different complexes in which these isoforms are envolved (Figure 7).”
- Lines 728-731, 736-738: “Transcription has emerged as an essential mechanism of TAD formation. Therefore, HP1 proteins in conjunction with architectural proteins at these sites, or with the basal transcription machinery at regions in which transcription occurs, could function in facilitating TAD formation and maintenance [71]…. The possible role of HP1a in the formation of TADs has not been studied, yet heterochromatin will play an essential role in maintaining these higher-order interactions.” Since authors did not find HP1a co-localization with TAD borders, currently there is no any support for this idea. Therefore, these sentences are completely speculative and should be deleted.
- Lines 733-735: “Most importantly, the maintenance of these territories and the established pericentromeric contacts also influence DNA-rich regions in the genome.” Change to: “Most importantly, the maintenance of these territories and the established pericentromeric contacts also influence active genome regions.”
- Lines 777-780: All GSE numbers in Supplementary Table 1 should be accompanied by the references on the corresponding works.
Reviewer 2 Report
I am satisfied with the author's revision addressing my concerns.
Author Response
We would like to thank the reviewer for taking the time to revise our manuscript and for all the suggestions which helped us improve the quality of our manuscript.
Round 2
Reviewer 1 Report
After revision, the manuscript of Meyer-Nava et al. became improved and after minor text editing listed below could be published in Cells.
Line 155: “Table 1. HP1a associated factors whose protein interacting domains have been identified” Change to: “Table 1. Proteins or factors in mammals and Drosophila that were revealed as directly bound with the known domains of HP1a/HP1α”
Lines 259-261: “In Table 2, we list the HP1a interactors — not necessarily direct interactions —defined only in Drosophila melanogaster, for these proteins, the exact domain (s) of interaction with HP1a have not been characterized.” Change to: “In Table 2, we list the HP1a interactors in Drosophila melanogaster — not necessarily direct interactors — for which the exact domain (s) of interaction within HP1a have not been characterized.”
Line 263: “Table 2. HP1a interacting proteins in which the interaction domains have not been identified” Change to: “Table 2. HP1a interactors in Drosophila for which the interaction domains within HP1a have not been identified”.
Author Response
Please see the attachment.

This manuscript is a resubmission of an earlier submission. The following is a list of the peer review reports and author responses from that submission.
Round 1
Reviewer 1 Report
Title: Heterochromatin, the dark side of the Force
Authors: Silvia Meyer-Nava et al.
- In this review paper, authors described the discovery history of heterochromatin protein (HP1) family, HP1 structure organization and features, HP1 evolutionary comparison among different species, HP1 subcellular localization and binding to chromosomes (heterochromatin and euchromotin), HP1 function and interaction with other chromatin proteins, in which there were also some domain-dependent analyses through bioinformatics by authors own included. It is a systematic and detailed summary.
- The author's research for many years mainly focuses on the fruit fly (Drosophila melanogaster) and therefore for the most part of this article the introduction of heterochromatin and HP1 function came out of Drosophila However, more readers for this article are working on the field of mammalian cell research, especially in humans and disease, including cancer. For this reason, it is recommended to complement the following contents.
- At the beginning of the article, the author described the heterochromatins in Drosophila cells in two paragraphs (rows32-48), which were brief and impressive. It is recommended for authors to come up with a similar introduction to the heterochromatins in mammalian cells, their quantity and distribution around the whole genome, so that readers have a general understanding to increase the importance of HP1 protein function. There can be no doubt that heterochromatins in mammalian cells are much more complex than those in Drosophila cells, since Drosophila has only four chromosomes, while human cell has 46 chromosomes. Therefore, it is more important for readers to have a general understanding about heterochromatin distribution in mammalian cells.
- In this paper, authors repeatedly discussed the inhibitory (silencing) role of HP1 on the expression of genes that are located in heterochromatin (rows 37-38, 50-56, 72-74, 103-105, 325-326, 333-337, 488-489, 498-501,595-599, etc.) and separately mentioned the enhance effect of HP1 on the expression of genes in euchromatin (rows 612-623.630-632), but there is a lack of practical examples. It is recommended for authors to provide specific examples of those genes whose expression is silenced, those whose expression is enhanced, including gene’s position and environment (promoters and enhancers), the mechanism of action of HP1, and which chromatin proteins or heterochromatic factors involved. It would be more beneficial to the reader's deep and vivid understanding of the function of HP1 protein family.
- I have two questions about the title.
- The content of this review mainly involves the function of HP1 protein family, why is it titled with “heterochromatin”.
- I don't quite understand why the first letter of the word Force is capitalized in the title, is there any special meaning there.
- An inaccuracy in the text is suggested to be corrected,
- All of “amino part or amino domain” (rows 88, 198, 412, 415, 534, 535) should be corrected to “amino-terminal part/domain or N-terminal part/domain”, which is the correct representation of the end of a polypeptide chain or a protein.
5. English language throughout the manuscript needs to be corrected and improved.
Author Response
We thank the reviewer for his/her revision. We hope this revised version of our manuscript is suitable for publication in Cells.
Please see the attachment.
Kind regards,
Viviana Valadez

Reviewer 2 Report
The review article of Meyer-Nava et al. is devoted to description of interacting partners of the main heterochromatin protein HP1a. I was not able to read the whole text of the manuscript since English language throught the text is very imperfect. It is hard to catch the sense of many sentences. So, I do not recommend to publish this review in Cells.
Below, are several examples of very long and/or grammatically incorrect sentences in the part of the text which I have read:
P.2: “One of the proteins found this way was is the Heterochromatin Protein 1 (HP1) which is a highly conserved protein [18] that was discovered initially in Drosophila by the group of Grigliatti in a study in which they found more than 50 loci that acted as suppressors of position effect variegation (PEV).”
P.2: “Mutations which affect HP1 proteins activities have a significant impact on organism development, for example in Drosophila the null mutants for HP1a are lethal before they can reach the end of embryonic stages [25] and although the isoforms are very structurally similar, they have different functions since one does not rescue the other.”
P.2: “Although the chromodomain is conserved in many other proteins, there are ways in which the cell indicates specificity for the regions where they must bind and carry out their functions, for example, when phosphorylation of histone H3 in serine 10 is immediately adjacent to the HP1 binding site at H3K9me, this binding is disrupted, and the protein does not remain bound [30], [31].”
P.3: “This domain conserves, a critical function for the formation of heterochromatin [39] which is the dimerization of HP1 proteins, and also contacts with many proteins that own a conserved pentapeptide motif, PxVxL (x=any amino acid) (Table 1) [40], [41].”
P.5: “It is thus possible that even a heterochromatin environment enriched with these modifications and proteins is dynamic, this is important because it has changed the view of heterochromatin from stable and rigid regions to regions which can also be highly malleable where diverse cellular mechanisms such as DNA repair or transcription can take place.”
P.6: “…which are ubiquity expressed…”
P.6: “So, if a protein is highly conserved during evolution, that means its function is of importance for life, so it was naturally selected to remain.”
P.6-7: “For example, HP1a is smaller than HP1b and c, and independently of the position of the CHD domain, HP1a presents quite like that in humans, between amino acids 20 and 80.”
P.7: “…it is possible that the negative regulators in which phosphorylate HP1α…”
P.7: “CTE sequences from these two parts of the proteins were analyzed, and no similarities between them or with any domain reported were found, but expressing them truncated and independent of the rest of the protein, these sequences are located in the cytoplasm, explaining the location in the nucleus and in the cytoplasm that is seen in the isoforms b and c [47].”
P.7: “The different places of the two main domains trough the three principal isoforms could explain their ability to bind more quickly or not, to specific proteins forming different complexes.”
P.7: “There is also another variant in this possible modulation equation of the function.”
Author Response
We thank the reviewer for his/her comments. We have addressed all the points raised, and have made substantial changes to the manuscript. We hope you will find this revised version suitable for publication in Cells.
Please see the attachment.
Kind regards,
Viviana Valadez

Reviewer 3 Report
The manuscript entitled “Heterochromatin, the dark side of the Force”, Meyer-Nava et al. provided an extended review on Heterochromatin Protein 1 (HP1). In detail the authors analyze the data on the structure of this protein, describe its functions and specific interactions with other proteins and RNA. Moreover, they underlined a possible role of HP1 in regulation of genes’ activity in euchromatin. The authors did a great job, and, in my opinion, this review currently provides the most complete information on HP1. There is no doubt that the review will be of interest to the scientific community.
In fact, I have the only significant comment on this work that concerns its title. I believe that the title of the review is much broader than its content and does not reflect the essence of the work. It seems to me extremely important to emphasize in the title that the study is dedicated to this particular protein, HP1.
The text is well written, and information is presented in a clear and concise fashion, but I have some technical comments:
- Repeatedly in the text the authors introduce the same abbreviations:
- 45, 481 – MBT,
- 61, 65 – PEV,
- 86, 109, 484 – CHD,
- 87, 118 – CSD,
- 353, 357 – APs.
This is somewhat confusing, especially since sometimes these abbreviations (e.g., MBT) are not used anywhere else in the text.
- At least one abbreviation was mentioned only once in the text:
- 45 – ZGA
- I found different spelling for ACF: l. 154, Table 2, l. 264.
- 62 – “… this way was is…”
- 262-269. Please, explain what did you looking for the degenerated motives LxVxL and CxVxL.
- Legend to Figure 1 is not informative. What is it on the left and on the right? What is the small pink element?
- 325 – An extra dot in the end on the subtitle.
- Please, check the text for double-spaces between words. It seems that there are in l. 347, 368, 399, and many others.
- Please, put a bracket after the “(vtd)” in l. 349.
- Please, put a comma before “and” in l. 451.
- Please, put a dot after the reference “[229]” in l. 623.
- Please, correct this part of the sentence in l. 665 – “all HP 1isoforms”. Should be “all HP1 isoforms”.
- 685 – “in vivo” should be in italic.
- It would be very useful for readers to see references to the figures and tables in subsection 9 of the manuscript.
- I understand that authors formed the list of references automatically, but it contains a lot of problematic points (I point out only some of them):
- Ref. 11 (L. 764-766) – there is no information about journal (as well as for ref. 15) and “running title” inside.
- Ref. 13 (l. 769-771) and 17 (l. 779-780) – types.
Author Response
We thank the reviewer for his/her suggestions. We have addressed all the points raised. We hope you find this version suitable for publication in Cells.
Please see the attachment.
Kind regards,
Viviana Valadez

Round 2
Reviewer 2 Report
In my previous review, I wrote that I was not able to read the whole text of manuscript of Meyer-Nava et al. due to imperfect English which did not alow me to catch the sense of many sentences. In the response letter, authors claimed that the article has undergone English language editing by MDPI. Now the text became more readable, although it is still not easy to do this. Since two other Reviewers rated the manuscript quite high, I decided to read the manuscript throughout to make up my own opinion. In fact, this manuscript represents the hybrid of the review and research article. The research part of the manuscript contains the authors’ i) in silico analysis of the presence of the PxVxL or similar motifs, involved in the HP1 binding, in the indirect HP1-interacting partners; ii) bioinformatics analysis of HP1a colocalization with the architectural proteins at the insulators, as well as at the TAD boundaries. The descriptive part of the review seems to me not interesting because some important works in the field remained not mentioned and not discussed, some statements are not supported by the references and their meaning remains unclear. However my major concerns are related to the research part of this review article.
- In Table 1 authors presented a list of proteins that either in pull-down or in two-hybrid system assays were shown to directly bind with the HP1. The list of potential HP1 interactors, for which there were no data showing their direct binding with the HP1, is presented in Table 2. Among the latter proteins, the authors have found only few proteins containing PxVxL and similar motifs responsible for direct binding with HP1. They concluded that “HP1a binding needs to be highly regulated; its function is very important to maintaining genome stability, so few proteins contain direct binding motifs.”(Line 308). However, the authors did not provide the percentage of proteins carrying these motifs in the list of direct HP1 interactors. Whether all of them contain these motifs? If not, it means that direct HP1 interactors have other motifs responsible for binding with HP1 and proteins frm the Table 2 may also possess these motifs. Any way, the conclusion that “few proteins contain direct binding motifs” may be not correct.
- The analysis of colocalization of HP1a with the architectural proteins was performed only at the level of visual colocalization of their ChIP-seq profiles in the Genome Browser (Fig. 4) and did not accompanied by any type of statistical analysis. As a result, the following statement (lines 459-461): “We observed that the HP1a peaks co-localize mainly with HIPP1 (42%), followed by Su (Hw) 28%, CP190 (26%), Mod(mdg4) 24%, CTCF (20%), GAF (16%), and Zw5 (13%)” does not provide any information about occasional or non-occasional nature of colocalization of those architectural proteins with HP1a because it does not contain the statistical analysis (for example, a permutation test).
- Based on their own analysis of publicly available data, the authors present on Fig. 5 the chromatin structure at the heterochromatin/euchromatin border in the 2R chromosome of Drosophila. However, the sharp decrease of HP1a and H3K9me2/3 mark at the border between constitutive heterochromatin and euchromatin was first described in the work of Riddle et al. (Genome Res. 2011, 21:147-63), not cited by the authors. On Fig. 5 the authors noted the lack of binding of CTCF and CP190 in the constitutive heterochromatin, but did not provide an explanation what does this mean.
- On Fig. 7, the authors tried to illustate their idea that “HP1a is also involved in the maintenance of higher-order structures in ways we do not yet understand.”(line 682). They compared publicly available ChIP-seq HP1a, RNA-PolII and H3K27ac profiles for Drosophila S2 cells with Hi-C map for these cells and claimed that HP1a “presents the same binding profile as transcriptionally active chromatin marks...However, HP1a is also found in intra-TADs..” (lines 679-681). From my point of view, the profiles for RNA-PolII and H3K27ac visually do not correlate with the HP1a profile. Moreover, HP1a is located not only in the active regions corresponding to TAD boundaries and inter-TADs, but also in the inactive regions inside TADs. In other words, there is no any correlation between HP1a presence and TAD boundaries. However, this analysis may be informative if the authors would perform the statistical analysis of colocalization, which was not yet done.
- The manuscript still contains many incorrectly formulated sentences. This is not only the problem of bad English. Rather, this is the authors' inability to articulate their thoughts clearly and perfectly.
In summary, I do not recommend to publish this review in Cells because it is not well written and contains many speculative conclusions, not supported by the facts.
Below, my text editing suggestions. However, my editing is limited to the first 6 pages of the text which is less than a quarter from the whole manuscript length. There are many more sentences awaiting improvement in the rest part of the manuscript.
Line 16: “Understanding these complexes will help us to clearly understand …” Change to: “Characterization of these complexes …”
Line 25: “Both kinds of proteins participate in maintaining the structure of DNA and regulating gene expression [3].” Change to: “Both kinds of proteins participate in maintaining the structure of chromatin and regulating gene expression [3].”
Line 37: “Staining different types of cells, Emil Heitz conceived the word heterochromatin more than 90 years ago, observing its retention of a more compact structure throughout the cell cycle [8].” Change to: “Staining different types of cells, Emil Heitz conceived the term “heterochromatin” more than 90 years ago, observing retention of its more compact structure throughout the cell cycle [8].”
Line 41-43: “During the 1960s, satellite sequences were identified, sequenced, and mapped to pericentromeric and telomeric regions of metaphase chromosomes and also at the periphery of interphase cells [10].” Change to: “During the 1960s, satellite sequences were identified, sequenced, and mapped to pericentromeric and telomeric regions of metaphase chromosomes located at the nuclear periphery of interphase cells [10].”
Line 50-52: “These mechanisms include DNA methylation, histone post-transcriptional modifications, histone deacetylation and chromatin binding proteins, and non-coding RNA and interference RNA pathways [11]–[13]. Embryonic stem cells in general, have less heterochromatin.” Change to: “These mechanisms include DNA methylation, histone post-transcriptional modifications, histone deacetylation, binding of chromatin proteins and non-coding RNA and RNA interference pathways [11]–[13]. Embryonic stem cells in general have less heterochromatin than differentiated cells.”
Line 61: “Based on the cytological criteria, one-third of the genome of D. melanogaster is considered heterochromatin, including telomeres, pericentric regions, and chromosome 4 [14].” Change to: “Based on the cytological criteria, one-third of the genome of D. melanogaster including telomeres, pericentric regions, and chromosome 4 is considered as the heterochromatin [14].”
Lines 64-69: “As development and differentiation progress, heterochromatin regions become more abundant since each differentiated cell expresses only a specific selection of genes. In the early Drosophila, embryo heterochromatin is established only when the rapid nuclear divisions slow, and zygotic gene activation takes place at the Mid Blastula Transition (MBT) [15]. A fly has approximately 15,500 genes on its four chromosomes [16]. In the eye-antenna disc in stage L2, 9,020 genes are expressed, and by the time of stage L3, 8,134 genes are expressed [17], which indicates that heterochromatin becomes more abundant as the cell differentiates and expresses a specific set of genes.”
The difference between 9,020 and 8,134 active genes at these two stages does not seem very convincing to illustrate an idea that heterochromatin becomes more abundant in the coarse of cell differentiation. In fact, heterochromatinization in the terminally differentiated cells is mainly enhanced not because of less number of active genes at these stages, but for the stronger repression of inactive genes in order to avoid their inappropriate expression. This idea is supported by several recent publications (Riddle et al. Genome Res. 2011;21:147–63; Marshall and Brand Nat Commun. 2017;8:2271; Pindyurin et al. Epigenetics and Chromatin 2018;11(1):65). I recommend to include these references in the Review after words “As development and differentiation progress, heterochromatin regions become more abundant [] ...”.
Lines 84-87: “One of the proteins identified through this screening is Heterochromatin Protein 1 (HP1), which is a highly conserved protein [22] that was initially discovered initially in Drosophila by the group of Grigliatti in a study in which the authors found more than 50 loci that acted as suppressors of PEV.” Change to: “One of the proteins identified through this screening is Heterochromatin Protein 1 (HP1). It is a highly conserved protein [22] that was initially discovered in Drosophila by the group of Grigliatti in a study in which the authors found more than 50 loci that acted as suppressors of PEV.”
Line 112: “...divided by a changeable-length hinge region...” Change to: “…divided by the hinge region of variable length …”
Line 120: “… the histone H1.4 linker…” Change to: “… the linker histone H1.4 …”
Line 122: “HP1 has been considered a seal of repression…” Change to “HP1 has been considered as a sign of repression…”
Line 125: “Su(var)2-502” change to “Su(var)2-502”
Line 126-129: “Furthermore, a significant reduction of HP1a near the centromeres and a decrease in survival past the third larval stage have been shown in flies that have a null allele of Su(var)2-5 and are trans-heterozygous for Su(var)2-502 [41].” Change to: “Furthermore, a significant reduction of HP1a near the centromeres and a decrease in survival till the third larval stage have been shown in flies that have a null allele of Su(var)2-5 and are trans-heterozygous for Su(var)2-502 [41].”
Line 145: ”This domain conserves a critical function for the formation of heterochromatin…” Change to: “This domain preserves a critical function for the formation of heterochromatin…”
Line 149: “For example, a single amino acid replacement inside the CSD (I161E) interrupts the dimerization of mouse HP1β [33].” Change to: “For example, a single amino acid replacement inside the CSD (I161E) prevents the dimerization of mouse HP1β [33].”
Line 150: “The absence of dimerization also triggers the loss of contacts with the nuclear factors of PxVxL motifs…” Change to: “The absence of dimerization also triggers the loss of contacts with the nuclear factors carrying PxVxL motifs…”
Line 166: “…because it is the most…” change to: “…because it contains the most...”
Line 199-201: “Flies also have two germ tissue-specific isoforms, HP1d and rhino, which are expressed in the ovaries and involved in silencing in the germline via Piwi-RNA clusters. HP1e also presents in the testes and is essential for paternal chromosome segregation through embryonic mitosis [122].” Change to: “Flies also have two germ line-specific isoforms, HP1d and rhino, which are expressed in the ovaries and involved in transposon silencing in the germline via piRNA clusters. HP1e is expressed in the testes and is essential for paternal chromosome segregation through embryonic mitosis [122].”
Line 234-239: “This competition leads to differences in the paralogs location and a gradient is observed in which HP1a or α in heterochromatic regions featuring potent DNA compaction and phase separation activities followed by HP1b or β in areas where there is a change from heterochromatin to euchromatin, the enrichment of HP1b or β works as a bridge, allowing the recruitment of gene activators which contribute to maintain open chromatin states and finally HP1c or γ in euchromatin areas with entirely different partners [71], [124], [129].” Divide into 3 sentences.